# A salt-free medium facilitating electrode prelithiation towards fast-charging and high-energy lithium-ion batteries

Yangtao Ou[1,4], Bao Zhang [2,4], Renming Zhan[1], Shiyu Liu[1], Wenyu Wang[1], Shuibin Tu[1], Yang Hu[1], Zihe Chen[1], Xiangrui Duan[1], Xiancheng Wang[1], Li Wang [3] & Yongming Sun [1] ✉

The substantial consumption of lithium ions and sluggish reaction kinetics at the anode detrimentally impact the deliverable energy and fast-charging capability of lithium-ion batteries with silicon-based anodes. The prevailing contact prelithiation method using an electrolyte medium can replenish the active lithium, but it may cause materials/electrode instability and bring barrier for lithium-ion transport. Here we explore a contact prelithiation methodology employing cyclic carbonate mediums that can enable spatially and temporally uniform prelithiation reaction. These mediums enable a delicate equilibrium between a lithium-ion diffusion and the intrinsic prelithiation reaction rate throughout the electrode depth. Not only does this prelithiation method serve the fundamental purpose of tackling lithium loss issue, but it also fosters the creation of a solid electrolyte interphase with favorable lithium-ion transport properties. By utilizing fluoroethylene carbonate as the medium for anode contact prelithiation, an Ah-level laminated Si/C‖LiCoO$_2$ pouch cell shows a significant enhancement in cell-level energy density by 42.7%. Moreover, a Si/C‖LiCoO$_2$ pouch cell achieves an 80.9% capacity utilization at a fast-charging rate of 10 C (6 min) and exhibits a low capacity decay rate of 0.047% per cycle. Such a prelithiation method demonstrates versatility across various cyclic carbonate mediums, electrodes, and scalability for industrial applications.

The quest for lithium-ion batteries (LIBs) with high fast-charging capability and high energy density has become a long-term endeavor for various applications, including portable electronics, electric vehicles, and smart grids[1–3]. Graphite anode is the predominant choice in prevalent LIBs. However, its capacity limit (372 mAh g$^{-1}$ in theory) poses a significant constraint on further enhancement of battery energy density. This limitation has prompted significant efforts towards the development of silicon (Si)-based anode, which offer higher specific capacity (3579 mAh g$^{-1}$ in theory) and a similar working potential[4–6].

However, the unavoidable lithium (Li) loss at the Si-based anode during the formation cycle (typically > 15%) results in a substantial decrease in battery energy density[7]. Moreover, Si-based anode often possesses sluggish electrochemical reaction kinetics, which becomes the primary cause for achieving high fast-charging capability of batteries[8,9]. In concert, these issues have substantially hindered the realization of desirable electrochemical performance of LIBs with Si-based anodes, considering all crucial parameters, including energy density, fast-charging capability, and cycle lifespan. To overcome

[1]Wuhan National Laboratory for Optoelectronics, Huazhong University of Science and Technology, Wuhan, China. [2]School of Materials and Energy, University of Electronic Science and Technology of China, Chengdu, China. [3]Institute of Nuclear and New Energy Technology, Tsinghua University, Beijing, China. [4]These authors contributed equally: Yangtao Ou, Bao Zhang. ✉e-mail: yongmingsun@hust.edu.cn

these limitations, considerable efforts have been independently directed towards compensating for the initial Li loss to augment the energy density, or towards enhancing the reaction kinetics of Si-based anode to improve the fast-charging capability of LIBs[10,11]. However, few studies have considered these issues in tandem to improve the overall electrochemical performance till now. While the Li consumption at Si-based anodes can be effectively offset by employing sacrificial Li metal through contact/electrochemical prelithiation strategies[12,13], these methods may pose potential hazards to the structural stability and electrochemical reaction kinetics of the Si-based anodes. Through the application of material structure design and electrolyte engineering, the fast-charging capability of Si-based anodes has been enhanced[14,15]. Nevertheless, the problem of initial irreversible Li consumption persists. Consequently, despite prior accomplishments in resolving specific individual issues, to date, few studies have focused on enhancing the overall electrochemical performance of silicon-based batteries, encompassing all aspects of energy density, fast-charging capability, and cycling lifespan. Thus, the quest for suitable strategies that can simultaneously addressing the aforementioned issues, represents the paramount challenge in realizing advanced LIBs with Si-based anodes.

Prelithiation is widely recognized as the most popular approach to compensate for the large initial Li loss and improve the energy density of LIBs[16,17]. Among the various prelithiation strategies, contact prelithiation of electrode employing Li metal as the Li source stands out as one of the most promising approaches for practical applications due to its highest prelithiation efficiency[18,19]. This approach capitalizes on the spontaneous electrochemical lithiation reaction between metallic Li and the electrode in the presence of an electrolyte medium with high Li$^+$ concentration (usually 1 molar L$^{-1}$, 1 M)[20,21]. However, it confronts substantial obstacles for practical implantation due to the

complex pre- or post-processing, the damage to the active material/electrode and harmful side product. As depicted in Fig. 1a, the prelithiation using high Li$^+$ concentration medium (regular electrolyte medium in literature) exhibits spatial nonuniformity across the electrode depth. Profound prelithiation occurs on the electrode surface that is in direct contact with the metallic Li, whereas the region distant from the surface layer undergoes significantly weaker prelithiation. This process could further induce volume change and stress disparities across the electrode depth, ultimately resulting in material and electrode damage. Moreover, due to the nonuniform prelithiation across the electrode depth, the side products of the prelithiation reaction exhibit an inhomogeneous distribution and predominantly accumulates on the electrode surface, which adversely impact Li$^+$ transport and subsequently attenuates the kinetics of the electrochemical reaction.

In this study, we elucidate the dependence of contact lithiation reaction kinetics on the properties of mediums, including Li$^+$ ion concentration and diffusion, and we uncover the regulation principle towards controllable and homogeneous contact prelithiation of electrodes. We reveal that an optimized concentration of Li$^+$ ions in the medium, typically around 10 millimolar (mM), can facilitate a balance between the Li$^+$ diffusion capability and the reaction rate of the material across the electrode thickness. We demonstrate that cyclic carbonate solvents, including fluoroethylene carbonate (FEC), propylene carbonate (PC), vinylene carbonate (VC), and a hybrid of ethylene carbonate (EC)/diethyl carbonate (DEC), can produce mM-level Li$^+$ ions in-situ through their reaction with metallic Li and thus function as effective mediums without the addition of Li salts. Moreover, we have successfully achieved uniform contact prelithiation of various battery electrodes, including Si/C anode and LiNi$_{0.6}$Co$_{0.2}$Mn$_{0.2}$O$_2$ (NCM622)

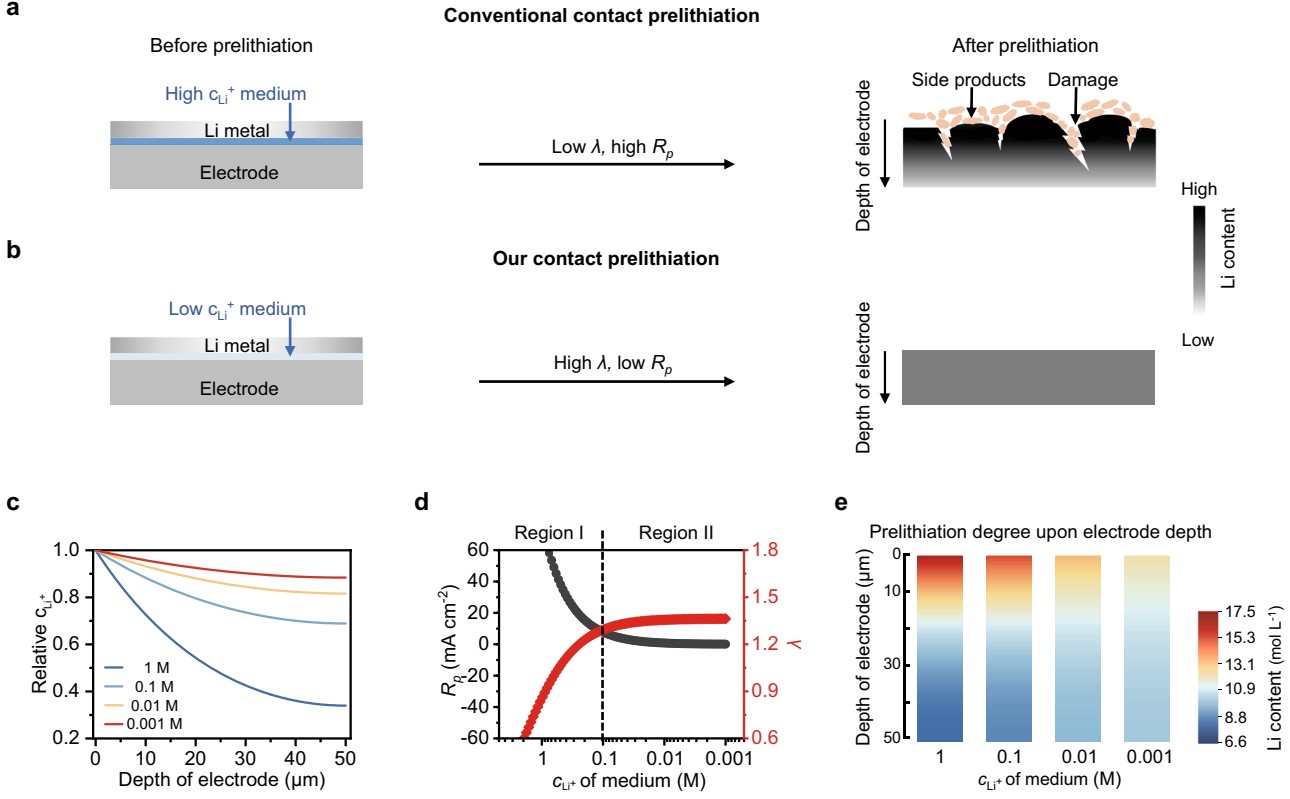

**Fig. 1 | Prelithiation protocols with different mediums. a, b** The effect of high (**a**) and low (**b**) $c_{Li^+}$ of mediums on electrode prelithiation. **c** Calculated relative $c_{Li^+}$ at different electrode depths in comparison to the superficial $c_{Li^+}$ at steady-state within the electrodes (5% prelithiation degree) employing mediums with different

$c_{Li^+}$. **d** $R_p$ and $\lambda$ as a function of $c_{Li^+}$. **e** Modulated Li distribution in Si electrodes by mediums with different $c_{Li^+}$. $R_p$ is defined as the prelithiation rate, $\lambda$ represents the diffusion characteristic parameter, and $c_{Li^+}$ signifies Li$^+$ concentration.

cathode, and the creation of a protective LiF-rich interphase using FEC medium. An Ah-level laminated pouch cell, employing a LiCoO$_2$ cathode and a prelithiated high-capacity Si/C anode (~1300 mAh g$^{-1}$), delivers a substantial increase in cell-level energy density (~42.7%) and increased cycling stability (93% capacity retention over 350 cycles at 1 C). Beyond the significantly improved electrochemical performance, such a salt-free medium circumvents the drawbacks associated with the conventional electrolyte medium, including Li salt residue, material/electrode damage, and complex post-processing. Consequently, our findings illuminate a promising pathway towards the realization of advanced LIBs with high energy and long lifespan.

## Results

### Prelithiation protocols with different mediums

We propose that the prelithiation medium with a low concentration of Li$^+$ ions can promote uniform and controllable prelithiation across the electrode depth and address the issues that high Li$^+$ concentration medium has, which thereby can facilitate the achievement of enhanced fast-charging capability and long-term electrochemical stability of LIBs (Fig. 1b). We then deeply reveal the underlying mechanisms for significant performance improvement using low Li$^+$ concentration medium for contact prelithiation. The contact prelithiation of the electrode encompasses the diffusion of Li$^+$ ions within the medium at the electrode, electron transfer in the electrode, and Li$^+$ ion-electron combination at the material side. Given the rapid electron transport capability, the rate of the prelithiation is determined by juxtaposing the diffusion rate of Li$^+$ ions within the medium and the combination rate of Li$^+$ ions and electrons at the materials, which is essentially the inherent lithiation reaction rate of the material. We calculate the steady-state Li$^+$ concentration distribution in the Si electrodes with a 5% prelithiation degree by prelithiation mediums with different Li$^+$ concentrations using PyBaMM[22], and the results showed that reducing Li$^+$ concentration in the medium can reduce the Li$^+$ concentration gradient across the electrode depth, and those with low Li$^+$ concentrations at millimolar (mM)-levels can exhibit a uniform distribution of Li$^+$ ions across the electrode depth, extending to tens of micrometers in depth (Fig. 1c). To elucidate the effect of Li$^+$ concentration during the electrode's prelithiation process, we further employed the prelithiation rate ($R_p$) to represent the material lithiation reaction rate, and the steady characteristic diffusion parameter ($\lambda$) of the porous electrode to indicate the supply of Li$^+$ ions from the medium (Fig. 1d). $R_p$ is dominated by the Li$^+$ concentration of the medium and the intrinsic lithiation reaction kinetics of the material, while $\lambda$ is mainly influenced by the effective diffusion coefficient and transference number of Li$^+$ ions. Taking Si anode as an example, the prelithiation process shows two characteristic distinct regions with the change of varying Li$^+$ concentrations. In the widely used high Li$^+$ concentration region (Region I, > 0.1 M), the $R_p$ can reach high current densities due to the significant driving force brought by direct contact with Li metal. Meanwhile, a small $\lambda$ is found due to the small effective Li$^+$ diffusion coefficient, leading to a pronounced Li$^+$ concentration gradient across the electrode depth. This suggests that employing the medium with high Li$^+$ concentration in Region I, the prelithiation process is spatially nonuniform across the electrode depth, which is accompanied by electrode/material stress and damage, as illustrated in Fig. 1a. Unexpectedly, the low Li$^+$ concentration region (Region II, < 0.1 M) presents a markedly different prelithiation behavior. In this region, $R_p$ is effectively reduced due to the significantly lower Li$^+$ concentration. Furthermore, the low Li$^+$ concentration medium, with its lower viscosity and higher Li$^+$ diffusion coefficient, results in a higher $\lambda$, leading to uniform Li$^+$ ions distribution over the electrode depth and mitigating the risk of particle and electrode damage from nonuniform prelithiation. The above simulation results are consistent with our experimental data (Supplementary Figs. 1–3 and Supplementary Table 1). These findings

suggest that the contact prelithiation process will benefit from low Li$^+$ concentrations of mediums.

The Li distribution across the electrode depth in mediums with different concentrations was modulated employing PyBaMM. As depicted in Fig. 1e, mediums with high Li$^+$ concentrations (typically, > 0.1 M) enable a substantial prelithiation degree gradient across the depth of the electrode. Moreover, nonuniform lithiation reaction with a much higher prelithiation degree at the surface than the inner part of the electrode leads to large local strain and thereby damage at the materials and electrode levels. Conversely, at lower Li$^+$ concentrations (<0.1 M), the prelithiation across the electrode depth is markedly more homogeneous.

### Characterizations of Si/C anode with FEC-mediated prelithiation

We reveal that dissoluble Li alkyl carbonates can be generated upon contact of cyclic carbonate with metallic Li, resulting in the production of Li$^+$ ions at a low concentration of ~7 mM (discussed latter). FEC, a cyclic carbonate, is utilized as the medium for contact prelithiation of high-capacity Si/C anode without the inclusion of additional Li salt, with the prelithiation product denoted as F-Si/C. Contrastingly, the product resulting from electrode prelithiation using a 1 M LiPF$_6$-based electrolyte is designated as E-Si/C. Figure 2a illustrates that the F-Si/C retains its initial morphology and structure in comparison to the pristine Si/C (refer to Supplementary Fig. 4), as opposed to the E-Si/C, which exhibits significant thickness variation (~60%) and microcrack formation. Therefore, it further corroborates the notion that a regular electrolyte medium (high Li$^+$ concentration) can inflict substantial damage to materials and electrodes due to the inhomogeneous prelithiation reactions, as depicted in Fig. 1a. To investigate the impact of a FEC medium on improving the spatial homogeneity of contact prelithiation across the electrode depth, the Li element contents of the prelithiation products at varying electrode depths were examined using augur electron spectroscopy (AES, Supplementary Fig. 5). The Li content exhibited no observable change from the surface to the bottom of the F-Si/C electrode, suggesting uniform prelithiation across the electrode depth[23] (Fig. 2b). Conversely, the Li content sharply decreased from the top surface to the bottom of the E-Si/C electrode, indicating spatial nonuniformity of prelithiation. Consequently, a low Li$^+$ concentration medium could address the issue of nonuniform, uncontrollable electrode prelithiation caused by regular electrolyte medium.

As depicted in the initial cycle charge/discharge profiles, the pristine Si/C electrode delivered a high discharge capacity of 1717 mAh g$^{-1}$ and charge capacity of 1334 mAh g$^{-1}$ at 0.1 C (0.1 A g$^{-1}$), resulting in a low initial Coulombic efficiency (ICE) of 78% (Fig. 2c). The fabricated F-Si/C and E-Si/C electrodes, both with a prelithiation degree of ~25%, exhibited elevated ICE of 109% and 86%, respectively. Different reversibility and stability of the two prelithiated electrodes could be attributed to the distinct prelithiation mediums employed (Supplementary Figs. 6 and 7). Notably, the F-Si/C electrode sustained a high capacity retention of 93% after 100 cycles at 0.5 C in half-cell configuration, significantly outperforming the Si/C and E-Si/C electrodes (91% and 65%, respectively). Therefore, FEC-mediated prelithiation (FM-prelithiation) could address the issue of low ICE without negatively impacting the electrochemical reversibility of the high-capacity Si/C electrode. The prelithiation degree of the F-Si/C electrode could be easily regulated by changing the reaction durations (Supplementary Fig. 8).

The interphase structure of the F-Si/C anode was characterized by X-ray photoelectron spectroscopy (XPS) and transmission electron microscopy (TEM). The results of XPS and TEM (Supplementary Figs. 9 and 10) verified that FM-prelithiation induced a robust preformed interphase that featured a remarkably higher proportion of inorganic species, and uniform, thin thickness compared to that of EM-prelithiation. As depicted in Fig. 2d and Supplementary Fig. 11, the F-Si/

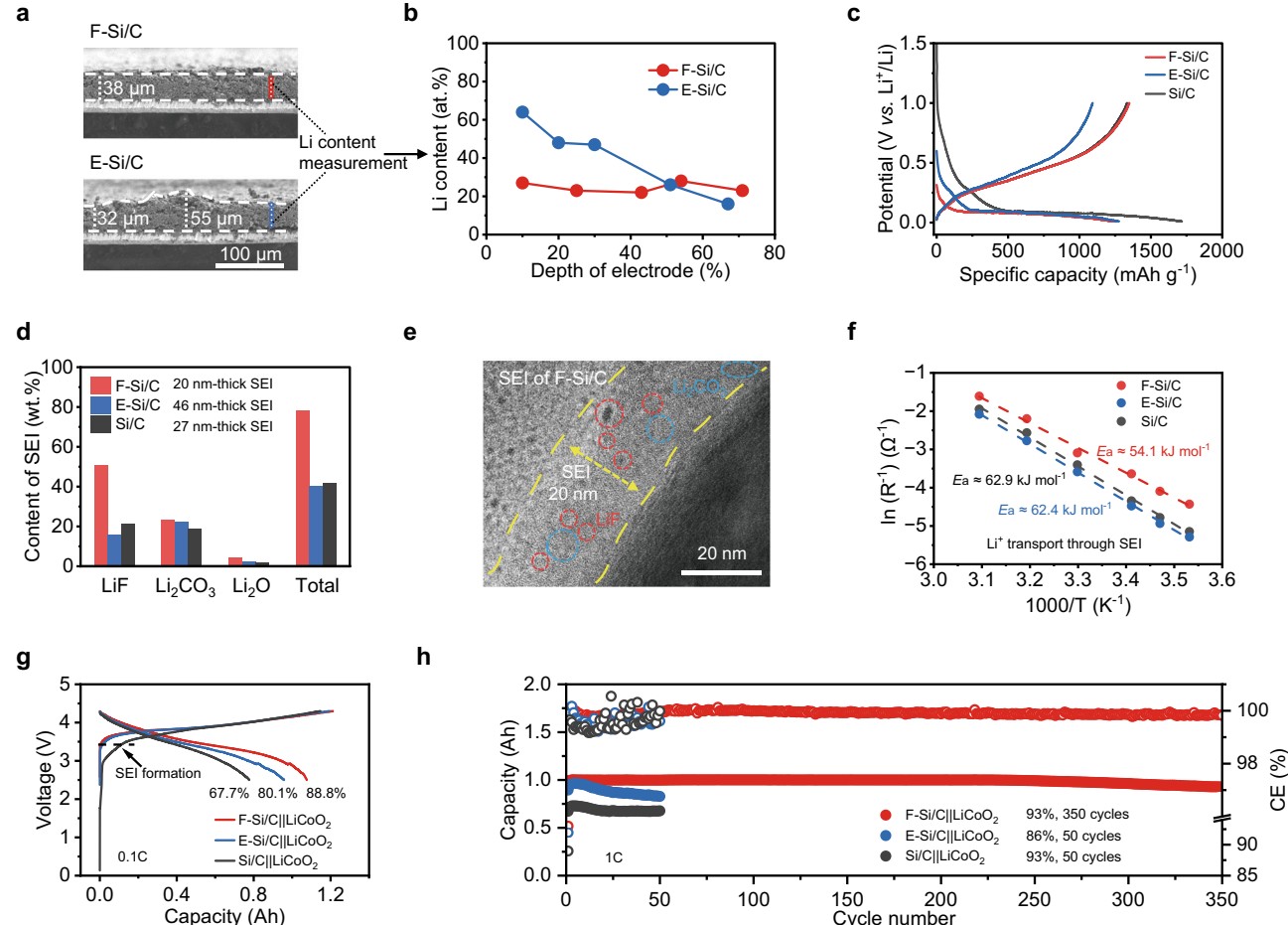

**Fig. 2 | Characterizations of Si/C electrode with FEC-mediated prelithiation.**
**a** Cross-sectional SEM images of the F-Si/C and E-Si/C electrodes. **b** The Li atomic content-electrode depth plot of the F-Si/C and E-Si/C electrodes by AES with the test locations labeled in (**a**). **c** The first-cycle voltage profiles of the F-Si/C, E-Si/C, and Si/C electrodes. **d** The content analysis of various SEI components of the F-Si/C, E-Si/C, and Si/C electrodes after discharge/charge cycling by XPS measurement.

**e** High-resolution TEM image of the F-Si/C after discharge/charge cycling.
**f** Comparison of activation energy ($E_a$) for the F-Si/C, E-Si/C, and Si/C electrodes. **g**, **h** The first-cycle voltage profiles at 0.1 C (0.1 A) (**g**) and the corresponding cycling performance (**h**) of the 1.2 Ah pouch cells using the F-Si/C, E-Si/C, and Si/C anodes in conjunction with the LiCoO₂ cathode at 1 C (1 A). The capacity retention rate is based on the first-cycle at 1 C.

C demonstrated a significantly elevated mass contents of LiF and total inorganic species, with values of 50.5 wt.% and 78.0 wt.% respectively, far surpassing the corresponding values of the Si/C (21.2 wt.%, 41.9 wt.%) and E-Si/C (15.5 wt.%, 40.3 wt.%). This suggests that FM-prelithiation contributes to the significant increase of the inorganic components of solid electrolyte interphase (SEI). TEM measurement was conducted to further investigate the microstructure of SEI post the discharge/charge cycling. An interphase layer with a heterogeneous thickness from 19 to 27 nm was observed on the pristine Si/C surface (Supplementary Fig. 12), indicating the inhomogeneity of the SEI without electrode prelithiation. An interphase layer with increased thickness and fluctuation (from 31 to 46 nm) was shown on the E-Si/C surface, which could be attributed to the aggravated side reaction and the uneven evolution of SEI during EM-prelithiation. Interestingly, a thin SEI with a uniform thickness of ~20 nm was observed (Fig. 2e). It is noteworthy that the SEI of Si/C and E-Si/C displayed amorphous structures devoid of crystalline domains, attributable to the overall low content of inorganic species. Conversely, the SEI of F-Si/C exhibited uniformly distributed crystalline domains of ~3 nm, which distinctly exhibited lattice fringes for LiF and Li₂CO₃ (Supplementary Fig. 13).

The Li⁺ transport capability of different electrodes was further examined by temperature-dependent electrochemical impedence spectroscopy (EIS) measurements ranging from 10 to 50 °C (Fig. 2f,

Supplementary Fig. 14, and Supplementary Table 2), and the corresponding activation energy was calculated based on the classic Arrhenius law, according to the fitted EIS results[24]. The activation energy for Li⁺ diffusion through the SEI ($E_a$) of the F-Si/C (~54.1 kJ mol⁻¹) was found to be the lowest, compared to that of the Si/C (~62.9 kJ mol⁻¹) and E-Si/C (~62.4 kJ mol⁻¹). The calculated Li⁺ diffusion coefficients ($D_{Li^+}$) of the F-Si/C by the cyclic voltammetry (CV) measurement were higher than those of the Si/C and E-Si/C, suggesting enhanced electrochemical reaction kinetics of the F-Si/C (Supplementary Fig. 15). The reduced $E_a$ and increased $D_{Li^+}$ in the F-Si/C substantiated the rapid Li⁺ transport capability at the F-Si/C surface.

To demonstrate the potential practical application of the F-Si/C anode, Ah-level laminated pouch cells were assembled and subsequently evaluated employing lithium cobalt oxide (LiCoO₂) cathodes and Si/C anodes with different prelithiation mediums (Supplementary Fig. 16 and Supplementary Table 3). The F-Si/C‖LiCoO₂ pouch cell demonstrated a high reversible capacity of ~1.08 Ah at 0.1 C (0.1 A), as shown in Fig. 2g. This capacity was 38.5% higher than the counterpart with the pristine Si/C anode without prelithiation (Si/C‖LiCoO₂, ~0.78 Ah at 0.1 C) and 12.5% higher than the Si/C electrode with electrolyte-mediated prelithiation (E-Si/C‖LiCoO₂, ~0.96 Ah at 0.1 C). Notably, the cell-level energy density of the F-Si/C‖LiCoO₂ cell increase

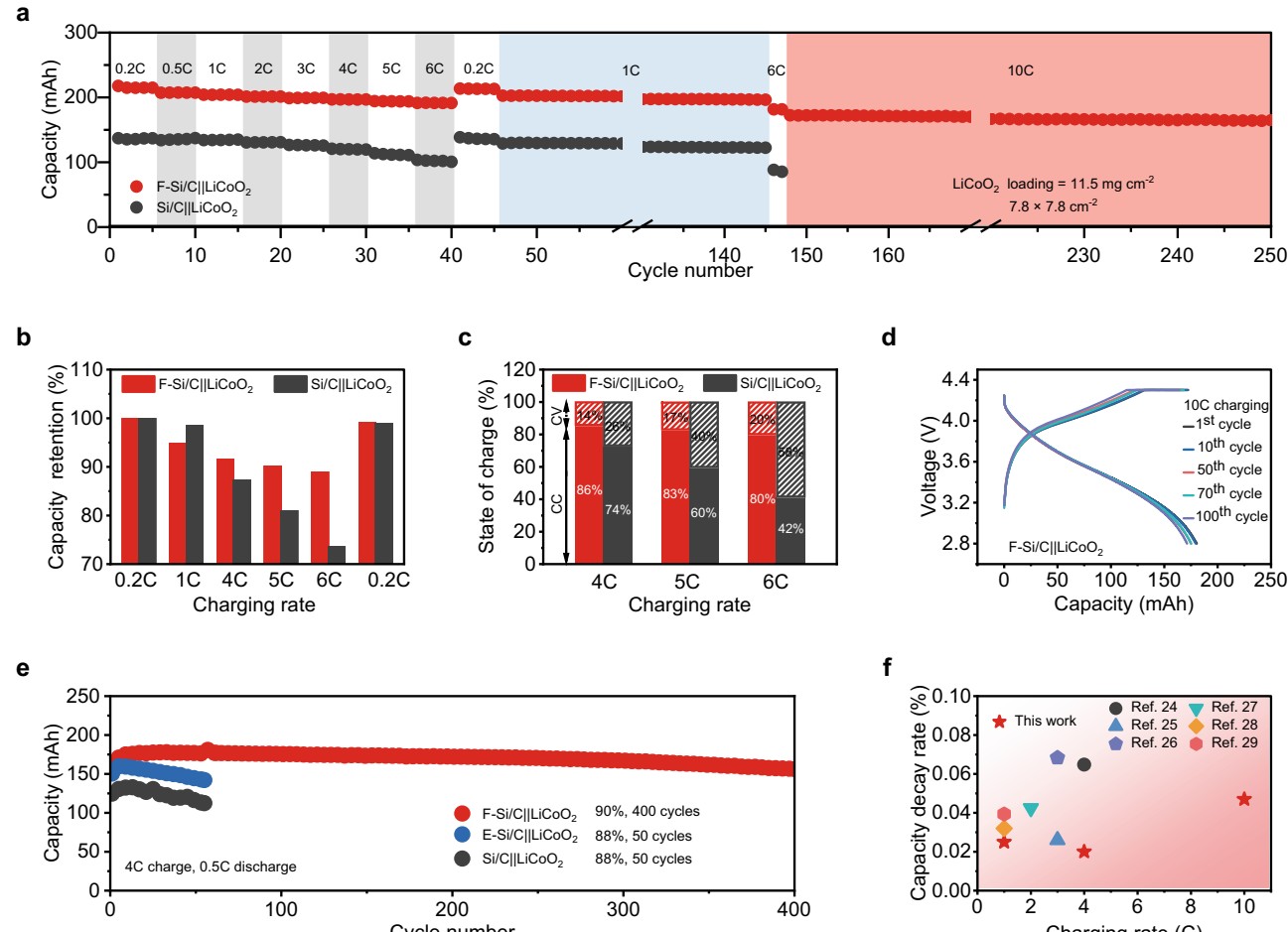

**Fig. 3 | Fast-charging capability of the Si/C anode with FEC-mediated pre-lithiation. a–d** Cycling performance of the 3-layer laminated F-Si/C‖LiCoO$_2$ and Si/C‖LiCoO$_2$ pouch cells with various charging rates ranging from 0.2 to 10 C (1 C, 240 mA), and discharging rate of 0.2 C (**a**), the corresponding comparison of capacity retentions (**b**), and plots illustrating the state of charge segmented into CC/CV stages at various charging current densities (**c**), and the voltage profiles of the F-Si/C‖LiCoO$_2$ cell during cycling with charging rate of 10 C (**d**). **e** Cycling performance of the F-Si/C‖LiCoO$_2$, E-Si/C‖LiCoO$_2$, and Si/C‖LiCoO$_2$ pouch cells with charging rate of 4 C and discharging rate of 0.5 C. **f** Comparison of capacity decay rate for long-term cycling with reported pouch cells with Si-based anodes at charging rate exceeding 1 C (refer to Supplementary Table 5).

reached 42.7% in comparison to the counterpart without prelithiation (464.6 *vs.* 325.7 W h kg$^{-1}$, Supplementary Fig. 17 and Supplementary Table 4). Highly overlapped voltage-capacity profiles were observed for the laminated F-Si/C‖LiCoO$_2$ pouch cell during long-term electrochemical cycling and the capacity retention of the cell reached 93% for 350 cycles at 1 C, outperforming the Si/C‖LiCoO$_2$ and E-Si/C‖LiCoO$_2$ cells (93% for 50 cycles and 86% for 50 cycles, respectively, Fig. 2h and Supplementary Fig. 18). Notably, the Ah-level laminated F-Si/C‖LiCoO$_2$ pouch cell achieved performance levels that exceed several recently published advanced pouch cells equipped with high-capacity Si-based anodes (>1000 mAh g$^{-1}$), in terms of overall cell capacity, energy density, and cycling lifespan (Supplementary Table 5). Thus, the employment of FEC medium for prelithiation successfully compensate for Li loss, increased the energy density and lifespan of the batteries. EIS and XPS measurements for the cycled Si/C, F-Si/C, and E-Si/C were further conducted (Supplementary Figs. 19 and 20). The results confirmed that FM-prelithiation could enable a robust and inorganic-rich SEI with low values for the R$_{SEI}$ and R$_{ct}$, thereby supporting its improved electrochemical reversibility. Due to the enhanced SEI stability, the F-Si/C‖LiCoO$_2$ cell exhibited better high-temperature storage performance compared to both the Si/C‖LiCoO$_2$ and E-Si/C‖LiCoO$_2$ cells (Supplementary Fig. 21).

## Fast-charging capability of the Si/C anode with FEC-mediated prelithiation

We evaluated the fast-charging capability of the F-Si/C anode using laminated F-Si/C‖LiCoO$_2$ pouch cell with a practical constant current–constant voltage (CC–CV) charging protocol, incorporating various charging rates from 0.2 to 6 C (5 h–10 min-charging rates, Fig. 3a). The long-term cycling stability under fast-charging rate of 10 *C* (6 min) was subsequently evaluated. The Si/C‖LiCoO$_2$ cell without anode prelithiation exhibited a recharge capacity of 87.3 and 73.5% (relative to the capacity at 0.2 C) within time intervals of 15 and 10 min (corresponding to charging rates of 4 and 6 C), respectively (Fig. 3b). Notably, the F-Si/C‖LiCoO$_2$ cell achieved high capacity retention of 91.5% and 89.0% at 4 and 6 C, respectively. Meanwhile, the F-Si/C‖LiCoO$_2$ cell demonstrated reduced potential polarization with much lower median voltage difference values compared to its counterpart without prelithiation under various charging rates from 0.2 to 6 C (Supplementary Fig. 22). Typically, the median voltage difference values under 6 C charging rate were determined to be 0.56 and 0.89 V for F-Si/C‖LiCoO$_2$ and Si/C‖LiCoO$_2$ cells, respectively. Correspondingly, cells employing the F-Si/C anode delivered a substantially higher percentage of charge capacity in the CC state compared to those with a pristine Si/C anode for all the test charging condition (Fig. 3c). Notably,

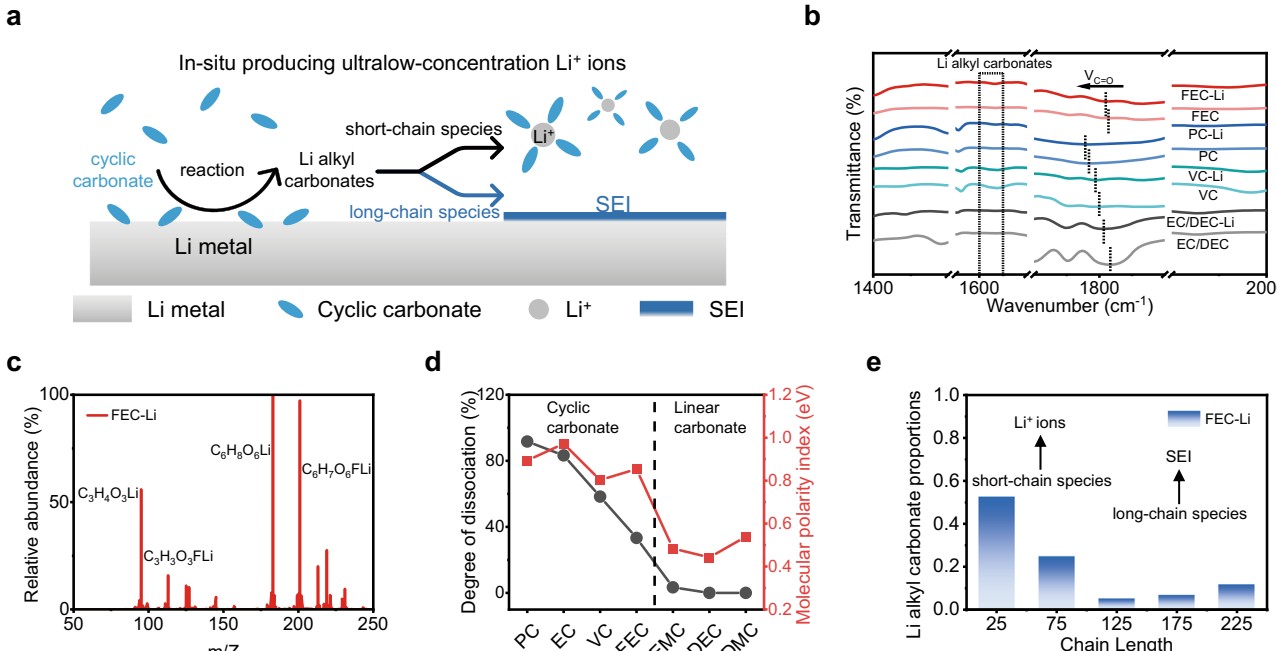

**Fig. 4 | Prelithiation mechanism for salt-free medium. a** Schematic of in-situ producing low-concentration Li⁺ ions via the reaction between Li metal and cyclic carbonate solvent. **b** FTIR spectra of various solvent-Li solutions. **c** Mass spectra of the FEC-Li solution. **d** Dissociation degrees of LiEC (a typical organic Li alkyl carbonate) in different solvents and the corresponding molecular polarity index, reflecting the solubility of LiEC. **e** Calculations of the structure of Li alkyl carbonates produced via the FEC-Li reaction. Metallic Li reacts with cyclic carbonate, resulting in the production of mM-level Li⁺ ions and passivated SEI.

the F-Si/C‖LiCoO$_2$ cell demonstrated a high capacity contribution of 86%, 83% and 80% in the CC state at charging rates of 4, 5 and 6 C, respectively. This performance significantly surpassed the mere contributions of 74%, 60% and 42% observed in the Si/C‖LiCoO$_2$ cell. Following the capacity measurement under various charging conditions, continuous cycling at charging rates of 1 and 10 C was conducted. The F-Si/C‖LiCoO$_2$ pouch cell demonstrated favorable cycling stability at fast-charging rate of 10 C. This configuration not only attained an 80.9% capacity filling in merely 6 min (relative to the capacity achieved at 0.2 C) and but also sustained a high capacity retention of 95.3% for 100 cycles. Highly overlapped voltage-capacity profiles for different cycles were shown under charging rate of 10 C, the cell displayed an ignorable increase in value for median voltage difference (0.032 V) for 100 cycles (Fig. 3d).

We then conducted the measurement of long-term cycling stability of the cells under fast-charging conditions. The F-Si/C‖LiCoO$_2$ cell exhibited a capacity retention of 90% after violent cycling for 400 cycles at charging rate of 4 C, underscoring its good electrochemical cycling stability in fast-charging scenarios (Fig. 3e and Supplementary Fig. 23). In contrast, the Si/C‖LiCoO$_2$ and E-Si/C‖LiCoO$_2$ cells displayed significantly lower capacity retention under same test conditions, maintaining only 88% for 50 cycles for both configurations. The enhanced electrode stability and suppressed Li plating behavior of the F-Si/C anode can be attributed to robust electrode structure and the fast Li⁺ transport capability, which significantly improve the fast-charging performance (Supplementary Figs. 24 and 25). Note that CC–CV fast-charging can result in a measured CE slightly above 100% due to measurement artifacts. To demonstrate the advantages of our solvent-mediated prelithiation technique in enhancing the fast-charging performance of high-energy-density LIBs with Si-based anodes, we executed a comprehensive comparison of the fast-charging performance, referencing results reported in existing literature (All data were collected from pouch cells with cycle number over 100). The F-Si/C‖LiCoO$_2$ cells exhibited a low capacity decay rate under

fast-charging conditions, with 0.025% for 1 C (350 cycles), 0.02% for 4 C (400 cycles) and 0.047% for 10 C (100 cycles), outperforming many of the reported pouch cells utilizing Si-based anodes[24–29] (Fig. 3f and Supplementary Table 5). The aforementioned analysis highlights the unparalleled role of solvent-mediated prelithiation in facilitating fast-charging, high-energy, and long-lasting LIBs.

**Prelithiation mechanism for salt-free medium**

Driven by its inherent reactivity, metallic Li initiates the reduction of carbonate solvents, resulting in the production of Li alkyl carbonates[30,31] (Fig. 4a). Among these, the short-chain variants dissolve in the solvent, leading to low-concentration Li⁺ ions, while the long-chain variants become the primary constituents of the SEI. The ability to establish a protective passivation layer on the Li surface, in conjunction with promoting the formation of soluble Li⁺ ions, constitutes a critical criterion for medium selection. We then measured the components and properties of different solvents post their reaction with metallic Li to investigate the reaction between cyclic carbonate and metallic Li, where FEC, PC, VC, and a hybrid of EC/DEC based solutions were designated as FEC-Li, PC-Li, VC-Li, and EC/DEC-Li solutions, respectively (Supplementary Fig. 26). FEC-Li, PC-Li, VC-Li, and EC/DEC-Li solutions displayed vibrations in Fourier transform infrared spectroscopy (FTIR) spectra at 1600–1650 cm⁻¹, which corresponded to the characteristics of organic Li alkyl carbonates (Fig. 4b), including ROCO$_2$Li and (ROCO$_2$Li)$_2$[32,33]. The FTIR vibrations of C = O shifted to lower wavenumbers compared to a pristine solvent reference, suggesting the interaction between the Li alkyl carbonate species and the solvents[34–36]. The existence of Li⁺ solvation structure for these solvent-Li solutions was also confirmed by the results of Raman measurement (Supplementary Fig. 27). The dissolved Li alkyl carbonate product in these solvent-Li solutions was further substantiated by the results of mass spectrometry and investigation. The high abundance of ions at m/z 95.032, m/z 113.022, m/z 183.047, and m/z 201.038 could be ascribed to the presence of C$_3$H$_4$O$_3$Li⁺, C$_3$H$_3$O$_3$FLi⁺, C$_6$H$_8$O$_6$Li⁺, and

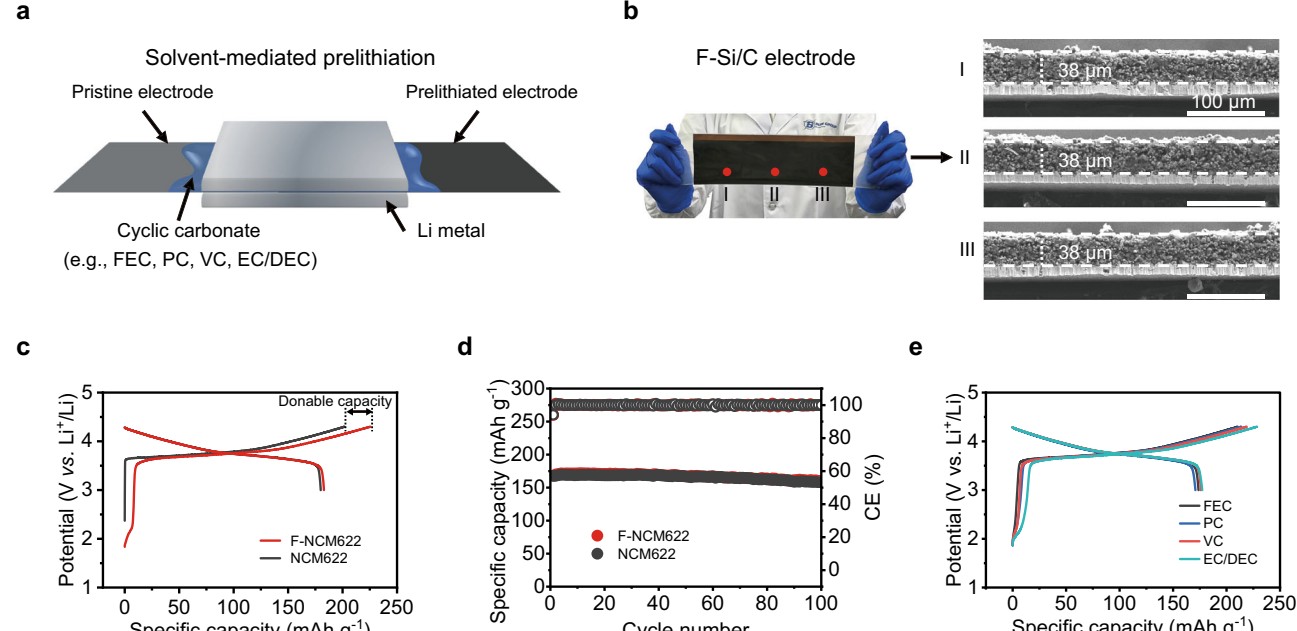

**Fig. 5 | Application to various electrodes and solvents. a** Schematic for solvent-mediated contact prelithiation process. **b** The photograph of the F-Si/C electrode and the cross-sectional SEM images taken from the selected location of the F-Si/C electrode. **c**, **d** The first-cycle voltage profiles at a specific current of 0.02 A g⁻¹ (**c**) and the cycling performance at a specific current of 0.1 A g⁻¹ (**d**) of the F-NCM622 and NCM622 electrodes. **e** The first-cycle charge voltage profiles at a specific current of 0.02 A g⁻¹ of the NCM622 electrodes after contact prelithiation for 30 min employing different cyclic carbonate solvents.

$C_6H_7O_6FLi^+$ in the FEC-Li solution[37,38] (Fig. 4c). The outcomes derived from nuclear magnetic resonance (NMR) measurements revealed a consistent Li⁺ concentration in FEC-Li following a specified resting period, which makes FEC as a stable prelithiation medium (Supplementary Figs. 28–30). Furthermore, the Li⁺ concentration in the FEC-Li solution was quantified (7 mM) using inductively coupled plasma optical emission spectrometry (ICP-OES) (Supplementary Fig. 31). The ionic conductivity values observed for a range of cyclic carbonate-Li solutions approximated those of the electrolyte with a low Li salt concentration, and were markedly lower than those of the standard electrolyte, as illustrated in Supplementary Fig. 32. Consequently, the low Li⁺ concentration solution, produced by the reaction between metallic Li and the solvent, could facilitate a mild reaction condition for contact prelithiation of the electrode. It was noteworthy that the reaction products between metallic Li and pure linear carbonate solvents (such as ethyl methyl carbonate, dimethyl carbonate, and DEC) were insoluble in the corresponding solvents, excluding their function as mediums for contact prelithiation (Supplementary Figs. 33–35).

To gain an in-depth insight into the mechanism for producing Li⁺ ions under the contact between Li metal and cyclic carbonate solvents, we undertook electrostatic potential calculations (Supplementary Fig. 36) and molecular dynamics (MD) simulations (Supplementary Fig. 37) using Li ethyl carbonate (LiEC), a typical organic Li alkyl carbonate. As illustrated in Fig. 4d, cyclic carbonates, with their elevated molecular polarity index (MPI), facilitate the dissociation of Li alkyl carbonate, thus conferring Li⁺ conductivity to the solvent, while the situation for linear carbonate solvents is inverse. It should be noted that the creation of a robust polymer film on the Li metal surface is essential for suppressing the continuous Li-solvent reaction. As shown in Fig. 4e and Supplementary Fig. 38, our calculation results of the self-produced Li alkyl carbonates reveal that while the long-chain Li alkyl carbonates are capable of forming an extensive polymer network in FEC and participating in the formation of a stable SEI on the Li metal surface, short-chain Li alkyl carbonates can dissolve into the solvent to produce Li⁺ ions. These findings validate the stable Li⁺ concentrations

in FEC-Li solution and effective passivation of Li metal in FEC, contributing to damage-free, uniform contact prelithiation.

## Application to various electrodes and solvents

As illustrated in Fig. 5a, a contact prelithiation method suitable for large-scale electrodes and adaptable to industrial processes, was explored, employing various cyclic carbonate-containing solvents as the mediums without the addition of Li salt, including FEC, PC, VC, and a hybrid of EC/DEC. Utilizing FEC as a typical solvent medium, Si/C electrodes with dimensions of 8 × 25 cm were successfully prelithiated (~25% prelithiation degree). Intriguingly, the electrode exhibited uniform structure, all the while preserving its initial intact morphology and integrity in different locations, as evidenced in Fig. 5b. In addition to the Si/C anode, a variety of other electrodes, including silicon oxide and graphite electrodes, also undergo successful prelithiation (Supplementary Figs. 39 and 40).

Contrary to graphite and Si-based anodes, intercalation-type cathodes demonstrate a heightened sensitivity in terms of structural integrity and stability upon contact with metallic Li. To demonstrate the universality of our electrode prelithiation approach, we fabricated a variety of prelithiated cathodes, including NCM622, LiFePO₄, and LiMn₂O₄, using a FEC medium. As depicted in Fig. 5c and Supplementary Fig. 41, the NCM622, LiFePO₄, and LiMn₂O₄ electrodes with ~10% prelithiation degree (F-NCM622, F-LiFePO₄, and F-LiMn₂O₄) concurrently exhibited reasonable donable capacities during their initial charge processes[39]. All the prelithiated cathodes delivered identical discharge capacities and working potential compared to their counterparts without prelithiation, showing their good stability. For instance, the initial charge capacities for the pristine NCM622 and F-NCM622 were 202 and 225 mAh g⁻¹ at a specific current of 0.02 A g⁻¹ with the cut-off voltage of 4.3 V (vs. Li⁺/Li), demonstrating a reasonable donable Li capacity of 23 mAh g⁻¹ for the F-NCM622 (Fig. 5c). During the discharge process, the F-NCM622 and NCM622 displayed similar capacities of 183 and 180 mAh g⁻¹, indicating that the FM-prelithiation of NCM622 did not result in structural damage to the bulk active

material. Furthermore, both the pristine and prelithiated NCM622 electrodes exhibited an average discharge voltage of 3.77 V and 3.76 V (*vs.* Li$^+$/Li), respectively. This, along with the highly overlapped CV curves (Supplementary Fig. 42), further supported the significant advantage of the solvent-mediated cathode prelithiation for achieving good electrochemical stability. F-NCM622 and NCM622 electrodes delivered initial capacities of 168.4 and 167.7 mAh g$^{-1}$ and capacity retention of 95.1% and 94.4% for 100 cycles at a specific current of 0.1 A g$^{-1}$, respectively, suggesting that the as-explored solvent-mediated prelithiation would not cause negative effect on the electrochemical cycling stability of the F-NCM622 cathode (Fig. 5d). The robust characteristics of the F-NCM622, F-LiFePO$_4$ and F-LiMn$_2$O$_4$ were further corroborated through a range of materials and electrochemical characterizations (Supplementary Figs. 43–47). The above results indicated the advantage of the solvent-mediated prelithiation for achieving good structural integrity and electrochemical stability.

We demonstrated that a variety of cyclic carbonate solvents can successfully prelithiate NCM622 electrode (Fig. 5e). The prelithiated NCM622 electrodes with different mediums displayed some deviations in capacity donation due to the difference in ionic conductivity of the solvent-Li mediums. The results showed that various cyclic carbonate-containing solvents, including FEC, PC, VC, and EC/DEC hybrid, could work as effective mediums for prelithiation due to the self-formation of Li$^+$ conductive Li alkyl carbonates via metallic Li-solvent reactions.

## Discussion

In conclusion, our study deeply reveals that low Li$^+$ concentration mediums can enable a delicate equilibrium between the Li$^+$ diffusion and the intrinsic reaction rate of the material across the electrode depth, enabling spatially and temporally uniform and manageable contact prelithiation. We elucidate that cyclic carbonate mediums can produce mM-level Li$^+$ ions through their spontaneous reaction with metallic Li. Building upon this knowledge, we have demonstrated that a FM-prelithiation strategy simultaneously introduces additional active Li, augments electrochemical reaction kinetics, and enhances the stability of the Si-based anode, thereby improving the comprehensive electrochemical performance of batteries, including energy density, fast-charging capability, and cycling lifespan. This efficient prelithiation method is demonstrated to be universally applicable for various anodes (Si/C, silicon oxide, and graphite), cathodes (NCM622, LiFePO$_4$, and LiMn$_2$O$_4$) and cyclic carbonate solvents (FEC, PC, VC, and EC/DEC hybrid), and shows its generality, scalability, and adaptability to industry processes. The solvent-mediated prelithiation circumvents the need for additional Li salt and the pre- or post-processing associated with the electrolyte in comparison to the existing prelithiation approach, offering a straightforward, clean, safe, universal, and effective approach for the industry, thereby catalyzing the advancement of high-energy LIBs.

## Methods
### Preparation of electrodes
Si/C slurry was made by mixing 80 wt% of commercial Si/C powder (Canrd), 10 wt% of carbon black (Canrd, super C45), and 10 wt% of lithium polyacrylate in deionized water (Canrd, 4% solid content). The slurry was then coated onto Cu foil (Canrd) homogeneously and dried at 80 °C in a vacuum oven for 20 h to fabricate the Si/C anode. The Si content of the Si/C powder was ~35 wt%, while the particle size averaged around 13 μm. To ensure the accuracy of the electrode fabrication, we followed a meticulous process using specialized equipment. Firstly, we employed an electric homogenizer (THINKY, ARM-310) to prepare a homogeneous slurry. Subsequently, an automatic coater (Kejing, MSK-AFA-I) equipped with a precision doctor blade (Kejing, EQ-Se-KTQ-150) with an accuracy of ±0.2 μm was utilized for stable and repeatable coating operations. Moreover, to maintain consistent

thickness throughout the electrode, we utilized an electric roller machine (Kejing, MSK-2150) known for its accuracy of ±1.5 μm. This ensured that the electrode coating thickness remained uniform across its entirety. Lastly, to verify the electrode thickness, measurements were taken at multiple points to calculate an average value, ensuring the desired level of precision and quality control. The thicknesses and the mass loading of the Si/C anode were 20 μm and 2 mg cm$^{-2}$, respectively. The LiCoO$_2$, NCM622, LiFePO$_4$, and LiMn$_2$O$_4$ electrodes on Al foil were fabricated using similar operations, which consisted of 90 wt% of active material, 5 wt% of carbon black, and 5 wt% of poly-vinylidene fluoride employing N-Methylpyrrolidone as the solvent for slurry. The mass loading of above cathode was 11.5, 10, 9.5, 9.5 mg cm$^{-2}$, respectively. The prelithiation of the electrodes was carried out through a contact operation between the pristine electrode and metallic Li, utilizing mediums including a carbonate solvent or electrolyte, for varying reaction times. This process was conducted in an Ar-filled glove box, with oxygen and water levels maintained at ≤ 0.5 ppm and ≤ 0.1 ppm, respectively.

For lab-scale electrode prelithiation fabrication, two glass plates are employed to secure the Si/C electrode (8 × 25 cm$^2$) and Li metal in intimate contact. During prelithiation, the two glass plates were subjected to a pressure of approximately 10 kPa, which effectively suppressed electrode rolling. Moreover, the overall degree of prelithiation in the electrodes was kept minimal (25% for the F-Si/C), thereby preventing significant strain that could lead to rolling.

### Materials characterizations
Scanning electron microscope (SEM) images were collected on a Gemini SEM 300 field-emission SEM instrument. X-ray diffraction measurements were conducted utilizing an Empyrean X-ray diffractometer equipped with Cu Kα radiation (40 kV, 30 mA, λ = 1.5418 Å). TEM measurements were performed with the FEI Tecnai G2 F30 instrument. FTIR were obtained using the Nicolet iS50R spectrometer. Raman spectroscopy measurement was executed on a Horiba LabRAM HR800 confocal Raman microspectrometer, employing 532 nm excitation. Elemental analysis was conducted via ICP-OES. XPS spectra were collected on an AXIS-ULTRA DLD-600W instrument. $^7$Li-NMR spectra were obtained on an Avance III 400 MHz NMR spectrometer. Mass spectrometry measurements were executed on a 1100 LC/MSD Trap. Lastly, AES measurements were conducted using the JEOL JAMP-9510F Auger electron microscopy to characterize the composition information at electrode depths.

### Electrochemical measurements
The electrochemical performance of the as-fabricated Si/C electrodes was examined in half-cell using coin-cell configuration (Canrd; CR2032 type), which employed metallic Li (diameter: 15.8 mm; thickness: 1 mm; manufacturers: China Energy Lithium Co., Ltd.) as the counter electrode, microporous polyethylene (Canrd; PP/PE/PP; porosity: 39%; average pore size: 0.028 μm; thickness: 25 μm) as the separator and 1 mol L$^{-1}$ (M) LiPF$_6$ in EC/DEC (1:1 v/v) with 5 vol% FEC, and 1 vol% VC additives as the electrolyte (100 μL, DoDoChem). The size of the Si/C electrode was 10 mm in diameter and these cells were assembled in an Ar-filled glovebox and tested at 25 ± 1 °C in ambient air. The ratio of the difference of discharge capacity between the fresh Si/C and the prelithiated Si/C and the discharge capacity of fresh Si/C was used to evaluate the prelithiation degree. EIS data was collected from 100 kHz to 100 mHz at the open-circuit potential with an amplitude of 10 mV. Before the EIS measurement, the system was allowed to stabilize at open-circuit voltage for 30 min. The EIS data analysis and scientific graphing were carried out using EC-LAB and Origin software, respectively. The cells were cycled three times at a 0.1 A g$^{-1}$ to ensure the stable formation of the SEI at 25 °C, and then EIS measurements were conducted at a series of gradient temperatures. Nyquist plot for spacer||PP||spacer cells with carbonate-Li solution was obtained and

the ionic conductivity was calculated by $\sigma = l/(R_b\,S)$, where $l$ was the thickness of the PP membrane (30 μm), $R_b$ was the resistance according to EIS, and $S$ was the disc-shaped spacer area (1.91 cm$^{-2}$). The CV was performed in the voltage range of 1–0.01 V at different sweep rates from 0.01 to 0.1 mV s$^{-1}$ using a Biologic VMP3 electrochemical workstation. The charge/discharge measurement of the Li‖Si/C and Li‖NCM622 cells was performed under 0.1 C for the initial three cycles and 0.5 C for the following cycles on a NEWARE battery tester instrument. The voltage range for the Li‖Si/C and Li‖NCM622 cells was 0.01–1 V and 3–4.3 V, respectively. The laminated F-Si/C‖LiCoO$_2$ pouch cells were evaluated at 0.1 C for the initial three cycles in the potential range of 4.3–2.5 V. The applied pressure for pouch cell during the electrochemical test was ~0.5 MPa. The dosage of electrolyte in the pouch cell was 3.5 g Ah$^{-1}$. For the 1.2 Ah cell (Size: 7.8 × 7.8 cm$^2$; Number of layers: 11; N/P ratio: 1.16), the current was set to 0.1 A for the initial three cycle and 1 A for subsequent cycles. For the 240 mAh cell (Size: 7.8 × 7.8 cm$^2$; Number of layers: 3; N/P ratio: 1.16), the current was set to 0.024 A for the initial three cycle and then the charging protocol combining CC–CV charging was optimized. The C rate was defined according to the practical capacity of the electrodes. The current densities for CC charging varied from 0.2 to 10 C (1 C = 240 mA). The corresponding total charging times (CC + CV) were fixed at 60, 30, 20, 15, 12, 10, and 6 min for 1, 2, 3, 4, 5, 6, and 10C-CV, respectively. The discharging current was fixed at 0.2 C. In the Li‖Si/C coin cell system, CE was determined as the ratio of the charging capacity to the discharging capacity during the corresponding cycling process, whereas in Li‖NCM622 coin cells and pouch cells, it is defined as the ratio of the discharging capacity to the charging capacity within each cycle. The specific capacity (mAh g$^{-1}$) was determined by calculating the ratio of the discharge capacity to the mass of the Si/C. The methodology for calculating energy density involved determining the ratio of discharge energy to the total mass of both cathode and anode, excluding the mass of the current collector.

## Calculation methods

The initial lithiation rate is determined by the balance between the reactions at the Li metal and Si electrode, expressed as:

$$i_0^{Li-ref}\frac{c_e}{c_e^{Li-ref}}\left(e^{\frac{0.5F\eta_{Li}}{RT}} - e^{\frac{-0.5F\eta_{Li}}{RT}}\right) = i_0^{Si-ref}\frac{3L\varepsilon_s}{r}\frac{c_e}{c_e^{Si-ref}}\left(e^{\frac{0.5F\eta_{Si}}{RT}} - e^{\frac{-0.5F\eta_{Si}}{RT}}\right)$$

$$\eta_{Li} = \phi_{Si} - \eta_{Si} \tag{1}$$

$$R_p = i_0^{Si-ref}\frac{3L\varepsilon_s}{r}\left(\frac{c_e}{c_e^{-ref}}\right)^{0.5}\left(e^{\frac{0.5F\eta_{Si}}{RT}} - e^{\frac{-0.5F\eta_{Si}}{RT}}\right)$$

Here, $i_O$ is the exchange current density, $c_e$ is the electrolyte concentration, $\eta$ is the overpotential, and $\varphi$ is the potential of the Si electrode, assumed to be 0.3 V vs. Li$^+$/Li. The exchange current densities for Li metal and Si are assumed to be 360 A m$^{-2}$ and 1 A m$^{-2}$, respectively, at a reference electrolyte concentration of 1 mol L$^{-1}$. $R$ and $T$ represent the ideal gas constant and temperature, respectively.

The Li$^+$ concentration distribution within the porous electrode is derived by solving the linear partial differential equation using PyBaMM package[22]:

$$\varepsilon_e\frac{\partial c_e}{\partial t} = \nabla(\varepsilon_e^{1.5}D\nabla c_e) + (1-t_+)\frac{j}{F}$$

$$j = i_0^{Si-ref}\frac{3L\varepsilon_s}{r}\frac{c_e}{c_e^{Si-ref}}(e^{\frac{0.5F\eta_{Si}}{RT}} - e^{\frac{-0.5F\eta_{Si}}{RT}}) \tag{2}$$

$$c_e(x=0) = c_e^s, \quad \frac{\partial c_e}{\partial x}\big|_{x=L} = 0$$

In this equation, $c_e^s$ is the electrolyte concentration at the electrode surface, $L$ is the electrode thickness (50 μm), $F$ is the Faradic constant, $D$ is the diffusion coefficient, $t_+$ is the cation transference number, $\varepsilon_s$ is the electrode phase volume fraction (0.6), and $\varepsilon_e$ is the electrolyte phase volume fraction (0.4). The diffusion coefficient and cation transference number are based on previous reports[40].

The steady-state concentration can be solved as[41]:

$$c(x) = c_e^s\frac{\cosh(\kappa_c(x-L))}{\cosh(\kappa_c L)}$$

$$\kappa_c = \left(\frac{3(1-t_+)\varepsilon_s i_0^{Si-ref}}{r\varepsilon_e^{1.5}FDc_e^{Si-ref}}\right)^{0.5}e^{\frac{F\eta_{Si}}{4RT}} \tag{3}$$

$$\lambda = \frac{1}{\kappa_c L}$$

The parameter $\lambda$ is crucial for the spatial concentration distribution, with a higher $\lambda$ indicating a more uniform distribution.

The electrostatic potential and molecular polarity index of various solvents were calculated using quantum chemistry calculations with the ORCA[42] program and Multiwfn[43] software. The SMD implicit solvation model and B3LYP/cc-pVTZ level of theory were employed. MD simulations were conducted using LAMMPS, with initial periodic systems prepared by PACKMOL[44] and Moltemplate (http://www.moltemplate.org/). The OPLS force field parameters, combined with RESP charges, were used. A concentration of 0.02 M LiEC was added to various solvents to simulate the dissolution of organic SEI components. The simulation protocol included Langevin dynamics at 500 K for 1 ns, followed by NPT and NVT runs at 300 K and 500 K, respectively. The solvation structures were analyzed from the last 2 ns of the trajectory, and the dissociation ratios for LiEC were determined by counting contact-ion-pair ratios. Additionally, 1 M LEC and LFEDC were added to DEC and FEC to model the Li alkyl component morphology. And VESTA[45] and VMD[46] were employed for visualizing the electrolyte structures.

## Data availability

Source data are provided with this paper. Source Data file related to theoretical calculations and simulations has been deposited in "MaterialsCloud" under accession code DOI link[47]. The data that support the plots within this paper and other finding of this study are available from the corresponding author upon request. Source data are provided with this paper.

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

## Acknowledgements

The authors thank the Analytical and Testing Center of Huazhong University of Science and Technology for providing the facilities to conduct the characterization. This work was financially supported by the Natural Science Foundation of China (Grant No. 52072137). The computational resources for this work were supported by Center for HPC, University of Electronic Science and Technology of China.

## Author contributions

Y.S. and Y.O. conceptualized the idea and oversaw the research. Y.O. performed the experiments and collected the data. R.Z., S.L., W.W., S.T., Y.H., Z.C., X.D., X.W., and L.W. contributed to the discussions. B.Z. executed the simulations and calculations. Y.S. and Y.O. undertook the data analysis and manuscript preparation, with contributions from all authors.

## Competing interests

The authors declare no competing interests.
