## [Transparent Peer Review file · Nature Communications]

A salt-free medium facilitating electrode prelithiation towards fast-charging and high-energy lithium-ion batteries

Corresponding Author: Professor Yongming Sun

Version 0:

Reviewer comments:

Reviewer #1

(Remarks to the Author)

This study reports Si/C powder prelithiated by employing cyclic carbonate mediums, which show improved electrochemical performance. The manuscript is well organized and some results are conducive to the development of high performance LIBs. The referee recommends major revision as below:

1. Whether the medium is consumed during the prelithiation? What is the selection criteria of medium besides the Li⁺ concentrations ?
2. What is the improvement of this method compared to previous reports which using chemical prelithiation and then immersing in electrolyte containing FEC? (Adv. Energy Sustainability Res. 2022, 3, 2200083) The novelty should be elaborated.
3. Why is the ICE of the E-Si/C electrode only 86% with capacity of 1000 mAh/g at the same prelithiation degree of F-Si/C? Besides, how to achieve the result of same prelithiation degree when the ICE is different.
4. What is the effect of FM prelithiation on the SEI? Why does it result in the formation of inorganic-rich and thin SEI?
5. The full cell of F-Si/C electrode displays much outstanding performance than other papers, the authors should clarify the reason.
- 6 . Considering the best performance of F-Si/C full cells, its expansion rate after cycling should be given, as well as the data for E-Si/C and pristine Si.
7. The stress analysis of F-Si/C and Si/C electrode is recommended to provide, in order to manifest the difference of their expansion.
8. The authors should give more details about the cell, such as electrolyte information and applied pressure during the test.
9. To clarify the cycling performance, the DCIR of different cycles for F-Si/C@ E-Si/C@Si full cells are also recommended to provide.
10. High temperature storage performance of the full cell for different electrode should be given in order to verify the stability of SEI and electrode with prelithiation.
11. The detail information of Si/C powder should be given, such as manufacturer, carbon content and particle size.
12. The area weight of anode is only 2 mg/cm each side with the thickness of 20 μm. How to control the accuracy of coating for these thin electrode?

Reviewer #2

(Remarks to the Author)

The manuscript presents an interesting contact prelithiation methodology utilizing pure cyclic carbonate mediums, which enables a spatially and temporally uniform prelithiation reaction. This approach significantly differs from traditional contact prelithiation methods that use electrolyte mediums with high Li salt concentrations. Using a Si-based anode as an example, the prelithiated Si/C electrode demonstrates markedly enhanced cycling stability and fast charging capability, along with effective Li compensation that increases the energy density of lithium-ion batteries. Besides, the authors thoroughly investigate the mechanism behind the enhanced electrochemical performance of the electrode post-prelithiation. Overall, the paper is well-structured and presents intriguing results. Therefore, I recommend acceptance of this work after addressing the following minor points.

1. The authors have demonstrated impressive cycling stability of the F-Si/C electrode, attributing the enhanced cycling

lifespan to the well-formed SEI. To provide more evidences for the robustness of the SEI, I recommend conducting more characterizations, including EIS and XPS measurements, throughout the electrochemical cycling process.

2. While the authors investigated the composition and structure of the SEI of the Si/C electrode, it is important to note that the surface structure of the prelithiated anode material may change after cycling. This suggests that the pre-formed SEI during the prelithiation process may evolve post-cycling. The authors should analyze the material interphase of the Si/C electrode after prelithiation and prior to cycling to capture these changes.

3. The authors underscore the markedly improved fast-charging performance of the F-Si/C electrode. However, they have not presented the anode structure/morphology following long-term cycling in a full cell configuration. A meticulous comparison between the cycled electrodes with and without the solvent-mediated prelithiation should be undertaken.

4. The authors assert that their contact prelithiation approach, which employs cyclic carbonate as the medium and metallic Li as the Li source, is effective for cathode prelithiation. Given the high chemical reactivity of Li metal, there is a potential for severe damage to the structure of the cathode material. The comparison of the NCM cathode with regular electrolyte mediated prelithiation and the cyclic carbonate mediated prelithiation should be carried out.

5. A significant finding in this study is that cyclic carbonate reacts with metallic Li, producing an ultralow Li-ion concentration solution, enabling uniform prelithiation of electrode. Do different electrodes exhibit variations in prelithiation efficiency, specifically, is there a difference in prelithiation capacity for different electrodes under the same operation time?

6. Experimental details about the pouch cell for cycling, such as the dosage of electrolyte in the pouch cell, and a clear methodology for energy density calculations, should be provided.

Reviewer #3

(Remarks to the Author)

It is a well-known fact that in LIB electrolytes, a thick-electrode generally requires a higher ionic conductivity in the electrolyte to achieve better performance, especially in terms of kinetic performance. However, the approach proposed in this paper claims uniformity in the depth direction of the electrode achieved with a very low-concentration condition, where ionic conductivity is inevitably low, which is hard to agree with. It is unclear how the calculation method used in Figure 1c was conducted and what assumptions were made to explain why there is a very low concentration gradient in a situation where the ionic conductivity must be low due to the extremely low concentration. If the explanation is that the viscosity of the electrolyte is reduced, then this is merely a computational effect, and in reality, the ionic conductivity would still be low, making it inconsistent with actual conditions. Furthermore, the prelithiation method used experimentally appears to involve fabricating a coin cell, performing prelithiation using a galvanostatic method, and then disassembling the cell to get a prelithiated electrode and to create a new one. (Even the detailed experimental procedure wasn't in the SI.) This method has been employed in numerous previous studies, making it neither novel nor practical. If the prelithiation was performed through direct contact with lithium metal in the presence of an electrolyte, it remains unclear how a low-ionic-conductivity electrolyte could ensure uniform prelithiation on both the electrode surface and its inside. The rationale for this claim appears insufficient. While Figure 5 presents a scale-up strategy, Figure 5a alone does not provide enough evidence to confirm whether a large-area electrode was indeed prelithiated using this approach. The use of FEC as a solvent is also not novel.

Above all, the methods proposed and used in this study are not novel at all, so I don't believe this paper is suitable for publication in the highly prestigious Nature Communications.

Reviewer #4

(Remarks to the Author)

This paper reported the uniform distribution of Li⁺ ion across the depth of electrode using various cyclic carbonate mediums during prelithiation, to boost the energy density, cycling performance improvements as well as fast charging capabilities. The author applied this method with various types of anode materials and paired with different cathodes. Moreover, comprehensive and thorough characterization techniques were implemented to elucidate the working mechanism. However, I have a few suggestions for the improvement and hope the author can find it helpful.

1. Definition of prelithiation degree and how to control? What is the prelithiation setup looks like? Normally, during prelithiation, the electron will roll up due to the volume change of Si, is there any pressure applied to the electrode during prelithiation in this study? If using roll-to-roll method as illustrated in Fig 5a, during long prelithiation process (~ 300 mins), how to prevent solvent evaporation?

2. Line 162 Auger electron spectroscopy (AES), what is the approx. depth of electrode in nanometer? There is bigger fluctuation of O species, what these contents could be?

3. Line 185-188, how did you normalize the wt% number for LiF and total inorganic species, and which signature peaks are considered as inorganic species?

4. What is the structure of pristine Si/C without any treatment? TEM data should be provided for comparison. Any lithiated Si/C compounds were identified in the bulk region after prelithiation?

Version 1:

Reviewer comments:

Reviewer #1

(Remarks to the Author)

The revised version is suitable for publication now.

Reviewer #2

(Remarks to the Author)

The authors have well addressed the comments so I would recommend publishing the manuscript as is.

Reviewer #3

(Remarks to the Author)

While I acknowledge that the results presented in this manuscript are technically impressive, I initially believed that the work lacked sufficient novelty to merit publication in a high-impact journal such as Nature Communications. However, in light of the editor's suggestion and the authors' responses, I have re-evaluated the manuscript and offer the following comments and questions

1. As the authors are aware, electrodes, such as those shown in Figures 1a and 1b, are not flat but inherently porous. In such porous electrode architectures, when the proposed direct contact prelithiation method is applied, lithium metal comes into immediate contact with the topmost silicon particles. This results in extremely rapid, uncontrolled prelithiation at the surface, while the interior silicon (near the current collector) remains unreacted. Based on the manuscript, it appears that the simulation model does not account for this direct Li–Si contact at the top surface. Even if the authors mitigate concentration gradients by using a low-concentration electrolyte, they cannot prevent over-lithiation of silicon particles that are directly touching lithium metal. Therefore, I argue that depth-wise nonuniformity in prelithiation is inevitable with this direct contact method, regardless of electrolyte engineering. From the standpoint of achieving uniform prelithiation across all particles in the electrode, I believe that this strategy is not the best method. If the authors are proposing a resting step (or self-discharge between the Si particles having different SOC) as a solution to this nonuniformity, I would like to ask how this is fundamentally different from what occurs under lithium metal contact with conventional electrolytes.
2. The authors replied that using FEC in the method of direct contact prelithiation is novel. Even if that is technically true (which I am not convinced of), FEC has long been known as a No.1. critical additive for forming stable SEI layers on silicon or silicon-carbon composite anodes. Thus, it is difficult to regard the use of FEC itself as a meaningful novelty.
3. The authors demonstrate excellent performance using various pouch cells, including a 1 Ah-level cell. While fabricating such a large cell at the university level is certainly not trivial, what ultimately matters is the electrodes' areal loading level. If the reported high capacity is primarily achieved by stacking many low-loading electrodes, it is not a practical approach and does not represent a high energy-density design. According to the table, the cathode loading appears to be ~ 2 mAh/cm², which is not considered high. In this case, the anode thickness must be extremely thin due to the high specific capacity of silicon. Could this be why the uniformity issue I raised in Comment 1 was not significant in your experiments? In other words, did the thinness of the electrode simply mask the depth-wise nonuniformity in prelithiation? If the proposed prelithiation method is truly practical and scalable, its effectiveness should be demonstrated in electrodes with at least 4 mAh/cm² areal capacity, and the full-cell tests should be conducted accordingly. The important point is not just achieving high Ah capacity, but doing so with industrially relevant electrode specifications.

Reviewer #4

(Remarks to the Author)

The authors have addressed my comments, I think the paper is ready to publish.

Version 2:

Reviewer comments:

Reviewer #3

(Remarks to the Author)

The authors have addressed the comments, so I would recommend publishing the manuscript.

Response to reviewers' comments:

(*Blue italic: Reviewers' remarks*; Black type: Our response)

Reviewer 1:

Comments:

This study reports Si/C powder prelithiated by employing cyclic carbonate mediums, which show improved electrochemical performance. The manuscript is well organized and some results are conducive to the development of high performance LIBs. The referee recommends major revision as below:

Response: Thank you very much for the reviewer's positive and encouraging evaluation of our work. To address the reviewer's concerns, a serial of additional experiments and discussion have been conducted during the revision. Please see the detailed responses to the following comments.

1. Whether the medium is consumed during the prelithiation? What is the selection criteria of medium besides the Li^+ concentrations?

Response: Thanks for your careful comments.

Solvents will be partially consumed during the prelithiation process. The interaction of solvents with Li metal results in the formation of both soluble and insoluble alkyl lithium carbonates. The insoluble alkyl lithium carbonates grow on Li metal and suppresses the continuous reaction. The reaction mechanism between metallic Li and carbonate solvent was exhibited in Fig. 4a.

The selection criteria of medium besides the Li^+ concentrations: the medium should engage in a reaction with Li metal, generating solid-phase products that seamlessly coat the Li metal surface, forming a protective passivation layer to inhibit ongoing reactions. Additionally, the reaction product comprises species that contain dissolved Li^+ and facilitate Li conductivity.

Fig. 4 | a, Schematic of in-situ producing ultralow-concentration Li^+ ions via the reaction between Li metal and cyclic carbonate solvent.

The following statements have been added to the Main text:

" The ability to establish a protective passivation layer on the Li surface, in conjunction with promoting the formation of soluble Li^+ ions, constitutes a critical criterion for medium selection.

" (Line 320-322, Page 17, Main text)

2. What is the improvement of this method compared to previous reports which using chemical prelithiation and then immersing in electrolyte containing FEC? (Adv. Energy Sustainability Res. 2022, 3, 2200083) The novelty should be elaborated.

Response: Thanks for your careful comments. These two methods are different in principle, operation and effect.

- 1). The reference literature adopts a chemical prelithiation method and then immerses it in an electrolyte containing FEC. This method enhances the prelithiation efficiency and helps to form a stable artificial SEI. Our paper has developed a one-pot contact prelithiation method using cyclic carbonate medium. Our approach effectively addresses issues including inadequate prelithiation uniformity and the formation of non-uniform SEI on the material surface, which traditional contact prelithiation approaches face.
- 2). Our salt-free prelithiation method effectively addresses issues including inadequate prelithiation uniformity and the formation of non-uniform SEI on the material surface, which traditional contact prelithiation approaches face, and brings more technical advantages, as follows:

Reference literature	This method	Advantage
Multi-step method: first preparing a reaction solution by combining Li metal and biphenyl, then immersing the electrode in the solution for prelithiation, and finally rinsing the electrode with electrolyte.	One-step method: Li metal comes into contact with the electrode under solvent immersion.	Easy operation
Using Li metal, biphenyl, DME, and electrolyte containing Li salt.	Using Li metal and carbonate solvent.	Low cost
Biphenyl and Li salt residual on the electrode.	No Li salt residual.	No residue
Only for prelithiation of silicon oxygen anodes.	Universality for various anode and cathode prelithiation. Potential scalability	Scalability and universality

3. Why is the ICE of the E-Si/C electrode only 86% with capacity of 1000 mAh/g at the same prelithiation degree of F-Si/C? Besides, how to achieve the result of same prelithiation degree when the ICE is different.

Response: Thanks for your careful comments. The prelithiation degree, as defined in our study, was calculated by determining the ratio of the difference in discharge capacity between the fresh Si/C and the prelithiated Si/C to the discharge capacity of the fresh Si/C. This calculation method was consistent with that described in reference [Advanced Energy Materials **13**, 2203256 (2023)]. The relevant data of F-Si/C, E-Si/C and Si/C electrodes are listed as follows:

Sample	Discharge capacity (mAh g ⁻¹)	Charge capacity (mAh g ⁻¹)	ICE (%)	Prelithiation degree (%)
F-Si/C	1235	1349	109	~25
E-Si/C	1276	1092	86	~25
Si/C	1717	1334	78	0

For the E-Si/C electrode, the discharge capacity reached 1276 mAh g⁻¹, while the charge capacity was 1092 mAh g⁻¹. Therefore, the ICE was calculated to be 86%. Regarding the degree of prelithiation, it was determined as (1717 - 1276) / 1717, approximately 25%. The ICE of the E-Si/C electrode was only 86%, with a charging capacity of ~1000 mAh g⁻¹, which was lower than the corresponding values of the F-Si/C electrode. Regarding to the difference in ICE values of E-Si/C and F-Si/C at the same prelithiation degree, the reason should be the distinct characteristics of the

SEI formed on the two electrodes. In the case of the E-Si/C electrode, the prelithiation process through the electrolyte medium led to non-uniform Li insertion. This non-uniformity triggered the formation of a relatively thick and uneven SEI layer (Figure R1). The thick and inhomogeneous nature of this SEI layer created a high-resistance barrier (Figure R2). A significant amount of Li^+ ions got trapped within this SEI layer during the initial charge-discharge cycles. As a consequence, less Li^+ was available for reversible electrochemical reactions, leading to a reduction in the reversible specific capacity and the ICE. This was the reason why the E-Si/C electrode exhibits a lower ICE of 86% compared to the 109% of the F-Si/C electrode, even when they had a comparable prelithiation degree. The phenomenon has been widely confirmed in previous publications [*ACS Applied Materials & Interfaces* **12**, 27017–27028 (2020); *Advanced Materials* **35**, 2300002 (2023)]. By meticulously controlling the experimental conditions, we effectively regulated the prelithiation process to ensure that Si/C anodes operating in different medium exhibited similar discharge capacities. This consistency indicates that an equivalent degree of prelithiation was achieved, irrespective of differences in ICE.

Figure R1. High-resolution TEM images for the E-Si/C (a) and F-Si/C (b).

Figure R2. (a) EIS spectra of the F-Si/C and E-Si/C electrodes before cycling and (b) the comparison of the corresponding impedance values.

4. What is the effect of FM prelithiation on the SEI? Why does it result in the formation of inorganic-rich and thin SEI?

Response: Thanks for your careful comments.

FEC-mediated (FM) prelithiation would influence the components and structure of SEI, induced a robust pre-formed interphase that featured a remarkably higher proportion of LiF species, and uniform, thin thickness compared to that of electrolyte-mediated prelithiation, which thereby affected the structural stability of SEI in the subsequent electrochemical cycling. To address the

reviewer's concerns, we conducted a detailed measurements and analysis of the XPS and TEM of the F-Si/C and E-Si/C.

- 1). The interphase structure of the F-Si/C anode was characterized by XPS measurement (Supplementary Fig. 9). The high-resolution F 1s XPS spectrum displayed a prominent peak at 685 eV, corresponding to LiF, which was attributed to the decomposition of FEC. FM prelithiation was observed to be advantageous for the formation of an initial interphase, which contained a remarkably higher proportion of LiF species compared to that of the electrolyte-mediated (EM) prelithiation.
- 2). TEM images revealed a uniform interphase layer with an ultrathin thickness of ~ 11 nm, which was in stark contrast to the uneven interphase with thickness ranging from ~ 28 nm to ~ 42 nm (Supplementary Fig. 10).

FM prelithiation formed a thin LiF-rich interface layer at the Si/C anode interface and contributed the even evolution of SEI. Such a pre-formed SEI would undoubtedly influence the post-formed SEI during cycling. The construction of uniform, thin LiF-rich layer could preferentially induce the decomposition of LiPF_6 to produce an inorganic-rich and thin SEI, thereby inhibiting the subsequent decomposition of the electrolyte. The phenomenon has been widely confirmed in previous publications [*Energy Storage Materials* **63**, 102996 (2023); *Nano Energy* **139**, 110951(2025); *Journal of Colloid and Interface Science* **680**, 364–372 (2025); *ACS Applied Materials & Interfaces* **12**, 18465–18472 (2020)].

The following figures and corresponding discussions were added to the Supporting information:

Supplementary Fig. 9 | XPS spectra of the F-Si/C and E-Si/C electrodes. a-e, High-resolution C 1s (a), F 1s (b), O 1s (c) and Li 1s (d) XPS spectra and survey XPS spectra (e) of the F-Si/C and E-Si/C electrodes. f, The corresponding content analysis of various SEI components.

" Based on XPS analysis, a preformed SEI was observed on the F-Si/C electrode, displaying a markedly higher proportion of inorganic species—namely LiF, Li₂CO₃, and Li₂O—relative to that on the E-Si/C electrode. " (Page 16, Supporting information)

Supplementary Fig. 10 | High-resolution TEM images for the E-Si/C (a) and F-Si/C (b).

The following statements have been added to the Main text:

" The results of XPS and TEM (Supplementary Figs. 9 and 10) verified that FM-prelithiation induced a robust pre-formed interphase that featured a remarkably higher proportion of inorganic species, and uniform, thin thickness compared to that of EM-prelithiation. " (Line 185-187, Page 9, Main text)

5. The full cell of F-Si/C electrode displays much outstanding performance than other papers, the authors should clarify the reason.

Response: Thanks for your careful comments. The reasons for the observed performance improvement are elucidated as follows:

1) Uniform prelithiated electrode structure and robust SEI enhanced electrochemical reversibility. The F-Si/C has spatially and temporally uniform prelithiation at the electrode level (as shown in Figs. 2a and 2b). Moreover, the SEI formed on the F-Si/C electrode is highly robust. As shown in Fig. 2d, the proportion of LiF species in the SEI of F-Si/C is significantly higher than that of other materials. Fig. 2e reveals that the SEI has a uniform and thin thickness. These features enhance the structural stability of the F-Si/C during subsequent electrochemical cycling, thereby improving the electrochemical reversibility of the electrode.

2) The SEI has excellent Li⁺ transport properties, enhancing fast-charging capability. Experimental comparisons show that the activation energy (E_a) for Li⁺ diffusion through the SEI of the F-Si/C electrode is the lowest, ~ 54.1 kJ mol⁻¹, while those of the Si/C and E-Si/C electrodes are ~ 62.9 kJ mol⁻¹ and 62.4 kJ mol⁻¹, respectively (Fig. 2f). A lower activation energy indicates easier Li⁺ diffusion. Additionally, the calculated Li⁺ diffusion coefficients (D_{Li^+}) of the F-Si/C electrode are higher than those of the Si/C and E-Si/C electrodes (Supplementary Fig. 15), suggesting superior electrochemical reaction kinetics of the F-Si/C electrode. The lower E_a and higher D_{Li^+} confirm the rapid Li⁺ transport ability on the surface of the F-Si/C electrode, thus enhancing the anode's fast-charging capability.

6. Considering the best performance of F-Si/C full cells, its expansion rate after cycling should be given, as well as the data for E-Si/C and pristine Si.

Response: Thanks for your careful comments. To address the concerns raised by the reviewer, we utilized SEM measurement to evaluate the thickness of the F-Si/C, Si/C, and E-Si/C electrodes before and after 100 cycles with a charge current of 4 C. Then, we calculated the expansion ratio of these electrodes. As depicted in Figure R3 and Supplementary Fig. 24, the thickness of the F-Si/C electrode increased from 38 μm to 47 μm . The Si/C electrode exhibited a significant increase from 24 μm to 100 μm , while the E-Si/C electrode demonstrated an increase from 55 μm to 121 μm . The corresponding expansion ratios were calculated as 24%, 317%, and 120%, respectively (Figure R4). Importantly, the F-Si/C electrode displayed the lowest expansion ratio among the tested electrodes, providing further evidence of its structural stability.

Figure R3 Cross-sectional SEM images of the F-Si/C (a), Si/C (b), and E-Si/C (c) electrodes before cycling.

The following figures and corresponding discussions were added to the Supporting information:

Supplementary Fig. 24 | Cross-sectional SEM images of the F-Si/C (a), Si/C (b), and E-Si/C (c) electrodes after 100 cycles.

" The SEM image revealed the thickness of different Si/C electrodes after 100 cycles. The thickness of the F-Si/C electrode increased from 38 μm to 47 μm . The pristine Si/C electrode exhibited an increase from 24 μm to 100 μm , while the E-Si/C electrode showed an increase from 55 μm to 121 μm . The corresponding expansion ratio were calculated as 24%, 317%, and 120%, respectively. Notably, the F-Si/C electrode demonstrated the smallest expansion ratio among these electrodes, further confirming its structural stability. " (Page 31, Supporting information)

Figure R4. The comparison of expansion ratio among the F-Si/C, Si/C, and E-Si/C electrodes after 100 cycles with a charge current of 4 C.

7. The stress analysis of F-Si/C and Si/C electrode is recommended to provide, in order to manifest the difference of their expansion.

Response: Thank you for your careful comments.

As suggested, we conducted stress measurements on 0.2 Ah LiCoO₂||Si/C and LiCoO₂||F-Si/C pouch cells. A force sensor array film (G-PMS, SMITsense) was used to monitor the pressure during operation, which was equipped with precisely arranged force sensor arrays. To perform real-time pressure measurements, the prepared pouch cells were clamped between fixture devices with a set target initial external pressure. The force sensor array film was ingeniously placed between the stainless-steel wall and the pouch cell. Figure R5 shows the in-situ pressure data recorded for each battery during the cycling process. During the first charge, the stack pressure change (ΔP) of the LiCoO₂||Si/C battery with a charge current of 4 C significantly increased by ~ 19.6 kPa. After 30 cycles, ΔP continued to increase, with a cumulative ΔP reaching ~ 15.4 kPa. In contrast, the ΔP increase of the LiCoO₂||F-Si/C battery was smaller after the same number of cycles, being about ~ 11.9 kPa during the first charge and a cumulative increase of ~ 4.6 kPa after 30 cycles. This phenomenon indicates that the internal pressure of F-Si/C is lower, which is highly consistent with the low expansion rate shown in Figure R4.

Figure R5. In-situ pressure measurements of the LiCoO₂||Si/C (a, b) and LiCoO₂||F-Si/C cells (c, d).

8. The authors should give more details about the cell, such as electrolyte information and applied pressure during the test.

Response: Thanks for your careful comments. The additional details about the full pouch cell was provided as follows:

- 1). The electrolyte information is 1 mol L⁻¹ LiPF₆ in EC/DEC (1:1 v/v) with 5 vol% FEC, and 1 vol% VC additives as the electrolyte. This information has been presented in the supporting materials.
- 2). The applied pressure during the test is ~ 0.5 MPa.

The following discussions were added to the Supporting information:

" The applied pressure for pouch cell during the electrochemical test was ~ 0.5 MPa." (Page 5, Supporting information)

9. To clarify the cycling performance, the DCIR of different cycles for F-Si/C@ E-Si/C@Si full cells are also recommended to provide.

Response: Thanks for your careful comments. As suggested, the DCIR of the LiCoO₂||Si/C, LiCoO₂||F-Si/C and LiCoO₂||E-Si/C full cells was recorded using NEWARE battery tester instrument. The DCIR values of the LiCoO₂||Si/C, LiCoO₂||F-Si/C and LiCoO₂||E-Si/C cells at different cycles were shown in Figure R6. The DCIR values of the LiCoO₂||F-Si/C cell remain stable and at a relatively low level after different cycles, which contrasts sharply with those of the LiCoO₂||Si/C and LiCoO₂||E-Si/C cells. For instance, after 100 cycles, the DCIR of the LiCoO₂||F-Si/C cell was the lowest, at 32.7 Ω, while the DCIRs of the LiCoO₂||Si/C and LiCoO₂||E-Si/C cells were 37.8 Ω and 61.7 Ω, respectively. The result indicates that the LiCoO₂||F-Si/C cell has a stable and low DCIR during cycling, thereby supporting its excellent cycling performance.

Figure R6. The DCIR of different cycles for LiCoO₂||Si/C, LiCoO₂||F-Si/C, and LiCoO₂||E-Si/C full cells at 0.2 C.

10. High temperature storage performance of the full cell for different electrode should be given in order to verify the stability of SEI and electrode with prelithiation.

Response: Thanks for your careful comments.

As suggested, high-temperature storage performance tests for different electrodes were conducted at 55 °C. Prior to the measurement, the LiCoO₂||Si/C, LiCoO₂||F-Si/C, and LiCoO₂||E-Si/C cells were subjected to a pre-cycling process (with a potential range of 4.3-2.5 V at 0.1 C, at 25 °C), which facilitates the formation of the SEI. Subsequently, all the cells were charged to a voltage of 4.3 V at 0.5 C and subjected to a storage period of 6 days at 55 °C. Following this storage period, the cells were allowed to rest at 25 °C and discharged to a voltage of 2.5V in order to measure the corresponding capacity values, denoted as “Retained capacity”, which was utilized to evaluate the performance of the cells during high-temperature storage. As shown in Supplementary Fig. 21, when stored at 55 °C, the OCV of the LiCoO₂||F-Si/C battery was more stable, and its retained capacity and capacity retention rate (110.3 mAh g⁻¹ and 70.9% respectively) were higher than those of the LiCoO₂||Si/C battery (38.3 mAh g⁻¹ and 39.4% respectively) and the LiCoO₂||E-Si/C battery (62.4 mAh g⁻¹ and 45.7% respectively). Therefore, the LiCoO₂||F-Si/C battery exhibited better high-temperature storage performance than the counterparts.

The following figures and corresponding discussions were added to the Supporting information:

Supplementary Fig. 21 | High-temperature storage performance for LiCoO₂||Si/C, LiCoO₂||F-Si/C and LiCoO₂||E-Si/C cells. The charge profiles before exposure to high temperature (a), the OCV during storage at 50 °C for 6 days (b) and the discharge profiles at 25 °C (c) of the LiCoO₂||Si/C, LiCoO₂||F-Si/C and LiCoO₂||E-Si/C full cells, and the corresponding comparison capacity retention (d).

" High-temperature storage performance tests for different electrodes were conducted at 55 °C. Prior to the measurement, the LiCoO₂||Si/C, LiCoO₂||F-Si/C, and LiCoO₂||E-Si/C cells were subjected to a pre-cycling process (with a potential range of 4.3-2.5 V at 0.1 C, at 25 °C), which facilitated the formation of the SEI. Subsequently, all the cells were charged to a voltage of 4.3 V at 0.5 C and subjected to a storage period of 6 days at 55 °C. Following this storage period, the cells were allowed to rest at 25 °C and discharged to a voltage of 2.5 V in order to measure the corresponding capacity values, denoted as "Retained capacity", which was utilized to evaluate the performance of the cells during high-temperature storage. When stored at 55 °C, the OCV of the LiCoO₂||F-Si/C battery was more stable, and its retained capacity and capacity retention rate (110.3 mAh g⁻¹ and 70.9% respectively) were higher than those of the LiCoO₂||Si/C battery (38.3 mAh g⁻¹ and 39.4% respectively) and the LiCoO₂||E-Si/C battery (62.4 mAh g⁻¹ and 45.7% respectively). Therefore, the LiCoO₂||F-Si/C battery exhibited better high-temperature storage performance than the counterparts. " (Page 28, Supporting information)

The following statements have been added to the Main text:

" Due to the enhanced SEI stability, the LiCoO₂||F-Si/C cell exhibited better high-temperature storage performance compared to both the LiCoO₂||Si/C and LiCoO₂||E-Si/C cells (Supplementary Fig. 21). " (Line 242-244, Page 12, Main text)

11. The detail information of Si/C powder should be given, such as manufacturer, carbon content and particle size.

Response: Thanks for your careful comments. The Si/C powder utilized in this study was obtained from Canrd Company. The carbon content of the powder was approximately 65%, while the particle size averaged around 13 μm.

The following descriptions were added to the Supporting information:

" The carbon content of the Si/C powder was ~65 wt%, while the particle size averaged around 13 μm . " (Page 3, Supporting information)

12. The area weight of anode is only 2 mg/cm each side with the thickness of 20 μm . How to control the accuracy of coating for these thin electrode?

Response: Thanks for your careful comments.

To control the coating accuracy of thin electrodes, we implement a comprehensive electrode preparation process:

1. Slurry Preparation: We utilize an electric homogenizer to prepare a homogeneous slurry. This equipment (THINKY, ARM-310) ensures the uniform dispersion of all components in the slurry, which is crucial for achieving consistent coating quality.

2. Coating Process: An automatic coater equipped with a doctor blade (Kejing, EQ-Se-KTQ-150) boasting an accuracy of $\pm 0.2 \mu\text{m}$ is employed. The automatic coater (Kejing, MSK-AFA-I) provides a stable and repeatable coating operation. The use of a doctor blade with such high accuracy allows for precise control of the coating thickness. By adjusting the gap between the doctor blade and the substrate, we can accurately deposit the desired amount of slurry, minimizing thickness variations across the electrode surface.

3. Rolling Process: An electric roller machine (Kejing, MSK-2150) with an accuracy of $\pm 1.5 \mu\text{m}$ is utilized for the rolling process. During rolling, the electrode is compressed to a specific thickness. The high-precision electric roller press ensures that the pressure applied is uniform, resulting in a consistent thickness across the entire electrode.

4. Thickness Measurement: To guarantee that the final product meets the specified thickness requirements, we measure the thickness of the electrode at five different points using micrometer screw. These points are strategically selected to represent different areas of the electrode. After obtaining the measurements, we calculate the average thickness. This method helps to account for any minor variations in the coating and rolling processes, ensuring that the overall thickness of the electrode is within the acceptable tolerance range.

The following descriptions were added to the Supporting information:

" To ensure the accuracy of the electrode fabrication, we followed a meticulous process using specialized equipment. Firstly, we employed an electric homogenizer (THINKY, ARM-310) to prepare a homogeneous slurry. Subsequently, an automatic coater (Kejing, MSK-AFA-I) equipped with a precision doctor blade (Kejing, EQ-Se-KTQ-150) with an impressive accuracy of $\pm 0.2 \mu\text{m}$ was utilized for stable and repeatable coating operations. Moreover, to maintain consistent thickness throughout the electrode, we utilized an electric roller machine (Kejing, MSK-2150) known for its accuracy of $\pm 1.5 \mu\text{m}$. This ensured that the electrode coating thickness remained uniform across its entirety. Lastly, to verify the electrode thickness, measurements were taken at multiple points to calculate an average value, ensuring the desired level of precision and quality control. " (Page 3, Supporting information)

Reviewer 2:

Comments:

The manuscript presents an interesting contact prelithiation methodology utilizing pure cyclic carbonate mediums, which enables a spatially and temporally uniform prelithiation reaction. This approach significantly differs from traditional contact prelithiation methods that use electrolyte mediums with high Li salt concentrations. Using a Si-based anode as an example, the prelithiated Si/C electrode demonstrates markedly enhanced cycling stability and fast charging capability, along with effective Li compensation that increases the energy density of lithium-ion batteries. Besides, the authors thoroughly investigate the mechanism behind the enhanced electrochemical performance of the electrode post-prelithiation. Overall, the paper is well-structured and presents intriguing results. Therefore, I recommend acceptance of this work after addressing the following minor points.

Response: Thank you very much for your careful comments. We have revised the manuscript to improve the quality according to your helpful suggestions.

1. The authors have demonstrated impressive cycling stability of the F-Si/C electrode, attributing the enhanced cycling lifespan to the well-formed SEI. To provide more evidences for the robustness of the SEI, I recommend conducting more characterizations, including EIS and XPS measurements, throughout the electrochemical cycling process.

Response: Thanks for your careful comments. As suggested, EIS measurements for the Si/C, F-Si/C and E-Si/C electrodes after different cycles were conducted, and the EIS results were shown in Supplementary Fig. 19. The values of the SEI resistance (R_{SEI}) and the charge transfer resistance (R_{ct}) for the F-Si/C electrode remained stable and low values after different cycles, in sharp contrast to the counterparts of the Si/C and E-Si/C electrodes. For instance, after 100 cycles, the R_{SEI} and R_{ct} of F-Si/C were 2 and 14 Ω respectively, which were lower than those of Si/C (8 and 27 Ω , respectively) and E-Si/C (41 and 45 Ω , respectively). The result indicated that the FM-prelithiation could enable a robust SEI with low values for the R_{SEI} and R_{ct} , thereby supporting its superior electrochemical reversibility.

To further investigate the characteristics of the SEI, XPS analysis was conducted on the Si/C, F-Si/C and E-Si/C electrodes after 100 cycles. In the high-resolution F 1s spectrum, peaks corresponding to LiP_xF_y and LiF were detected at 687 and 685 eV, respectively, for the Si/C and E-Si/C electrodes (Supplementary Fig. 20). The intensities of these two peaks were comparable, indicating that LiF was not the main component on the SEI of Si/C and E-Si/C. Of particular significance, the F-Si/C electrode exhibited a prominent peak corresponding to LiF, in stark contrast to the relatively weak peak observed for LiP_xF_y . This stark difference in peak intensities sets the F-Si/C electrode apart from the other electrodes studied. This implies that the LiF-rich SEI maintained its stability after long-term cycling. The well-formed SEI effectively inhibits electrolyte decomposition and enhances cycling stability.

The following figures were added to the Supporting information:

Supplementary Fig. 19 | EIS spectra of the prelithiated Si/C electrodes. a-b, EIS spectra of the Si/C, F-Si/C and E-Si/C electrodes after the 10th (a) and 100th (b) cycles. c, Comparison of the corresponding impedance values.

Supplementary Fig. 20 | XPS spectra of the Si/C, F-Si/C and E-Si/C electrodes after 100 cycles. a-e, High-resolution C 1s (a), F 1s (b), O 1s (c) and Li 1s (d) XPS spectra and survey XPS spectra (e) of the Si/C, F-Si/C and E-Si/C electrodes. f, The corresponding content analysis of various SEI components.

The following statements have been added to the Main text:

"EIS and XPS measurements for the cycled Si/C, F-Si/C and E-Si/C were further conducted (Supplementary Figs. 19 and 20). The results confirmed that FM-pretreatment could enable a robust and inorganic-rich SEI with low values for the R_{SEI} and R_{cb} , thereby supporting its superior electrochemical reversibility. " (Line 238-242, Page 12, Main text)

2. While the authors investigated the composition and structure of the SEI of the Si/C electrode, it is important to note that the surface structure of the prelithiated anode material may change after cycling. This suggests that the pre-formed SEI during the prelithiation process may evolve post-cycling. The authors should analyze the material interphase of the Si/C electrode after prelithiation and prior to cycling to capture these changes.

Response: Thank you for your detailed comments. In response to the concerns raised by the reviewers, we have performed a comprehensive analysis of the F-Si/C using XPS and TEM after prelithiation and prior to cycling.

1). The interface structure of the F-Si/C electrode was investigated by XPS measurement (Figure R7). The high-resolution F 1s XPS spectrum showed a distinct peak at 685 eV, corresponding to LiF, which was attributed to the decomposition of FEC. Thus, FM prelithiation was conducive to the formation of the LiF-rich interface layer.

2). TEM images revealed that the interface layer of F-Si/C was uniform and extremely thin (~11 nm) (Figure R8). Within this layer, nanodomains of LiF, Li_2CO_3 , and Li_2O —approximately 3 nm in size—were embedded. This observation indicated the formation of an inorganic-rich interfacial layer during the FEC-mediated contact prelithiation process.

In conclusion, FM prelithiation induced the formation of a robust pre-formed interface with a significantly higher proportion of LiF species and a uniform, thin thickness. These characteristics affected the structural stability of the SEI during subsequent electrochemical cycles. The uniform and LiF-rich pre-formed interface of F-Si/C contributed to the more robust SEI structure during cycling (Fig. 2d and 2e).

Figure R7. XPS spectra of the F-Si/C electrode. Survey XPS spectra (a) and high-resolution Li 1s XPS spectra (b), and the corresponding content analysis of various SEI components (c).

Figure R8. High-resolution TEM images for the F-Si/C.

3. The authors underscore the markedly improved fast-charging performance of the F-Si/C electrode. However, they have not presented the anode structure/morphology following long-term cycling in a full cell configuration. A meticulous comparison between the cycled electrodes with and without the solvent-mediated prelithiation should be undertaken.

Response: Thanks for your careful comments.

In response to your suggestion, we investigated the morphology and structure comparison of the F-Si/C electrode and pristine Si/C electrode after cycling in full pouch cell under fast charging.

As depicted in Supplementary Fig. 25, the digital and SEM images reveal noteworthy observations following 100 cycles at 4 C. The cycled pristine Si/C electrode exhibited substantial silvery-grey deposits (Supplementary Fig. 25a). In stark contrast, the cycled F-Si/C electrode displayed a conspicuous absence of such deposits (Supplementary Fig. 25c). Examination of the SEM images provided further insights into the nature of the electrodes. Supplementary Fig. 25b illustrated the presence of dendritic and mossy metallic Li structures on the protrusions of the cycled pristine Si/C electrode. In contrast, the SEM image of the cycled F-Si/C electrode (Supplementary Fig. 25d) showcased a smooth and clean surface devoid of any Li deposits. Notably, the F-Si/C electrode exhibited a uniform structure and experienced minimal changes in thickness (Figure R9a). In stark contrast, the Si/C electrode displayed a substantial increase in thickness accompanied by the emergence of numerous cracks (Figure R9b). These observations provided clear evidence that the F-Si/C electrode demonstrated reduced expansion and enhanced structural stability throughout the cycling process. The suppressed Li plating behavior and enhanced electrode stability of the F-Si/C electrode could be attributed to robust electrode structure and the enhanced Li^+ transport capability of the F-Si/C electrode, which significantly improve the fast-charging performance.

The following figures and corresponding discussions were added to the Supporting information:

Supplementary Fig. 25 | Optical photograph and SEM image of Si/C (a, b) and F-Si/C (c, d) anodes after 100 cycles with a charging current of 4 C.

" The morphology and structure of the F-Si/C electrode and pristine Si/C electrode after cycling in full pouch cell under fast charging were conducted. The digital and SEM images reveal noteworthy observations following 100 cycles at 4 C. The cycled pristine Si/C electrode exhibited substantial silvery-grey deposits (Supplementary Fig. 25a). In stark contrast, the cycled F-Si/C electrode displayed a conspicuous absence of such deposits (Supplementary Fig. 25c). Examination of the SEM images provided further insights into the structure of the electrodes. Supplementary Fig. 25b illustrates the presence of dendritic and mossy metallic Li structures on the protrusions of the cycled pristine Si/C electrode. In contrast, the SEM image of the cycled F-Si/C electrode (Supplementary Fig. 25d) showcases a smooth and clean surface devoid of any Li deposits. The suppressed Li plating behavior of the cycled F-Si/C electrode can be attributed to the enhanced Li⁺ transport capability of the F-Si/C electrode, which significantly improves the fast-charging performance." (Page 32, Supporting information)

Figure R9. Cross-sectional SEM images of the F-Si/C (a) and Si/C (b) after 100 cycles with a charging current of 4 C.

The following statements have been added to the Main text:

" The enhanced electrode stability and suppressed Li plating behavior of the F-Si/C anode can be attributed to robust electrode structure and the fast Li⁺ transport capability, which significantly improve the fast-charging performance (Supplementary Fig. 24 and 25)." (Line 292-295, Page 15, Main text)

4. The authors assert that their contact prelithiation approach, which employs cyclic carbonate as the medium and metallic Li as the Li source, is effective for cathode prelithiation. Given the high chemical reactivity of Li metal, there is a potential for severe damage to the structure of the cathode material. The comparison of the NCM cathode with regular electrolyte mediated prelithiation and the cyclic carbonate mediated prelithiation should be carried out.

Response: Thanks for your careful comments. To address the reviewer's concerns, XRD and SEM measurements were performed on the NCM cathode samples subjected to two different prelithiation methods: regular electrolyte-mediated prelithiation (referred to as E-NCM622) and cyclic carbonate-mediated prelithiation (referred to as F-NCM622).

1). The SEM analysis of the NCM622 and F-NCM622 samples demonstrated that the active particles subjected to FEC-mediated prelithiation retained their original structures without exhibiting any particle-level cracks (Supplementary Fig. 43). In contrast, the E-NCM622 electrode displayed fractured particles, indicating significant structural damage to the cathode material (Supplementary Fig. 45a).

2). The crystallinity of NCM622, F-NCM622 and E-NCM622 electrodes was investigated using XRD (Supplementary Figs. 44 and 45b). The NCM622 and F-NCM622 electrodes displayed three prominent characteristic peaks at 19° , 37° and 45° , which corresponded to the (003), (101) and (104) planes of NCM, respectively. In contrast, the XRD analysis revealed a notable decrease in the peak intensity for the E-NCM622 electrode, despite having undergone the same overall prelithiation level. These findings suggested that prelithiation through electrolyte-mediated contact resulted in non-uniform prelithiation products and compromised the crystalline structure of NCM622. Conversely, the results presented above highlight the significant advantage of FEC-mediated prelithiation in preserving the structural integrity of the active material.

The following figures and corresponding discussions were added to the Supporting information:

Supplementary Fig. 45 | SEM image (a) and XRD pattern (b) of the E-NCM622 electrodes.

" *Supplementary Fig. 45a showed the SEM image of the E-NCM622 electrode. Compared with the pristine NCM622 (Supplementary Figure 43a), it was clearly observed that the particles were broken. The E-NCM622 displayed a notable decrease in the XRD characteristic peak intensity in comparison to the pristine counterpart (Supplementary Fig. 45b and Fig. 44).* " (Page 54, Supporting information)

Supplementary Fig. 43 | SEM images of the NCM622 (a) and F-NCM622 (d) electrodes.

Supplementary Fig. 44 | XRD patterns of the NCM622 and F-NCM622 electrodes.

5. A significant finding in this study is that cyclic carbonate reacts with metallic Li, producing an ultralow Li-ion concentration solution, enabling uniform prelithiation of electrode. Do different

electrodes exhibit variations in prelithiation efficiency, specifically, is there a difference in prelithiation capacity for different electrodes under the same operation time?

Response: Thank you for your careful comments. To address the reviewer's concern, we compared the prelithiation capacities of various electrodes with prelithiation operation time for 1h, including Si/C, graphite, NCM622, and LiFePO₄). The electrode capacity was evaluated through electrochemical tests at 0.1 C. The prelithiation capacity was calculated by multiplying the capacity difference between the prelithiated electrode and the pristine electrode. As depicted in Figure R10, a 1-hour operation resulted in prelithiation capacities of 0.245 mAh for Si/C, 0.311 mAh for graphite, 0.260 mAh for NCM622, and 0.056 mAh for LiFePO₄ electrodes. These findings indicate variations in prelithiation capacities among different electrodes under identical operation times.

Figure R10. The prelithiation capacities of Si/C, graphite, NCM622, and LiFePO₄ electrodes employing FEC-mediated prelithiation with duration time of 1h.

6. Experimental details about the pouch cell for cycling, such as the dosage of electrolyte in the pouch cell, and a clear methodology for energy density calculations, should be provided.

Response: Thanks for your careful comments. The additional experimental details about the pouch cell is provided as follows:

- 1) The dosage of electrolyte in the pouch cell is 3.5 g Ah⁻¹.
- 2) The methodology for calculating energy density involves determining the ratio of discharge energy to the total mass of both cathode and anode, excluding the mass of the current collector.

The following descriptions were added to the Supporting information:

" The dosage of electrolyte in the pouch cell was 3.5 g Ah⁻¹. The methodology for calculating energy density involved determining the ratio of discharge energy to the total mass of both cathode and anode, excluding the mass of the current collector. " (Page 5, Supporting information)

Reviewer 3:

Comments:

1. It is a well-known fact that in LIB electrolytes, a thick-electrode generally requires a higher ionic conductivity in the electrolyte to achieve better performance, especially in terms of kinetic performance. However, the approach proposed in this paper claims uniformity in the depth direction of the electrode achieved with a very low-concentration condition, where ionic conductivity is inevitably low, which is hard to agree with.

Response: Thank you for your detailed comment.

1). We agree that thick electrodes require LIB electrolytes with high ionic conductivity to achieve good charge/discharge cycling, since electrolytes with high ionic conductivity can facilitate Li^+ diffusion across electrode thickness and thus improving the reaction kinetic. In this study, the low-concentration medium with low ionic conductivity is explored for contact prelithiation operation not for electrochemical charge/discharge cycling of LIBs. As described in the experimental section (Line 61, Page 4, Supporting information), all the electrochemical evaluations of the as-fabricated prelithiated electrodes for LIBs were paired with a conventional electrolyte (1 M LiPF_6 dissolved in EC/DEC (1:1 v/v) with 5 vol% FEC and 1 vol% VC) for charge/discharge cycling.

2). According to our calculation results using PyBaMM, reducing the concentration of Li^+ in the electrolyte can decrease the Li^+ concentration gradient in the depth direction of the electrode (Fig. 1c). In an electrolyte with a low concentration (< 0.1 M), the Li distribution in the depth direction of the electrode is significantly more uniform (Fig. 1e). We experimentally verified the Li distribution as shown in Figs. 2a and 2b. The prelithiated electrode exhibits a uniform electrode structure and a homogeneous Li distribution in the depth direction of the electrode. This work presents a low-concentration Li^+ medium (cyclic carbonate solvent) that can homogenize the electrode contact prelithiation process, which is the main contribution of this study. This innovative approach effectively addresses issues such as nonuniform prelithiation that traditional contact prelithiation methods face.

2. It is unclear how the calculation method used in Figure 1c was conducted and what assumptions were made to explain why there is a very low concentration gradient in a situation where the ionic conductivity must be low due to the extremely low concentration. If the explanation is that the viscosity of the electrolyte is reduced, then this is merely a computational effect, and in reality, the ionic conductivity would still be low, making it inconsistent with actual conditions.

Response: We appreciate your insightful questions.

1). **Calculation method:** We calculated the steady-state Li^+ concentration distribution in the Si electrodes with a 5% prelithiation degree by prelithiation mediums with different Li^+ concentrations using PyBaMM. The calculation method used in Fig. 1c is detailed in supporting information (Line 84-91, Page 5, Supporting information). The Li^+ concentration distribution within the porous electrode is derived by solving the linear partial differential equation using PyBaMM package:

$$\varepsilon_e \frac{\partial c_e}{\partial t} = \nabla(\varepsilon_e^{1.5} D \nabla c_e) + (1 - t_+) \frac{j}{F}$$

$$j = i_0^{Si-ref} \frac{3L\varepsilon_s}{r} \frac{c_e}{c_e^{Si-ref}} \left(e^{\frac{0.5F\eta_{Si}}{RT}} - e^{-\frac{0.5F\eta_{Si}}{RT}} \right)$$

$$c_e(x=0) = c_e^s, \quad \left. \frac{\partial c_e}{\partial x} \right|_{x=L} = 0$$

In this equation, c_e^s is the electrolyte concentration at the electrode surface, L is the electrode thickness (50 μm), F is the Faradic constant, D is the diffusion coefficient, t_+ is the cation transference number, ε_s is the electrode phase volume fraction (0.6), and ε_e is the electrolyte phase volume fraction (0.4). The diffusion coefficient and cation transference number are based on previous reports.

2). **Assumptions:** As noted, low-concentration electrolytes (<0.1 M) inherently exhibit low ionic conductivity. However, unlike conventional constant-current charging processes in full cells, the reaction kinetics during contact prelithiation are highly sensitive to electrolyte concentration. In this scenario, *the Li⁺ concentration gradient is governed by a reaction-diffusion balance rather than sole ionic conductivity.* To elucidate the effect of Li⁺ concentration during the electrode's prelithiation process, we employed prelithiation rate (R_p) to represent the material lithiation reaction rate, and the steady characteristic diffusion parameter (λ) of the porous electrode to indicate the supply of Li⁺ ions from the medium (Fig. 1c). In high-concentration electrolytes, rapid Li⁺ consumption (high R_p) exceeds the replenishment rate, creating steep concentration gradients. Conversely, *low-concentration electrolytes reduce R_p , while their lower viscosity and higher Li⁺ diffusion coefficient significantly improve λ .* Therefore, such a medium helps to achieve a delicate balance between Li⁺ diffusion and the intrinsic reaction rate of the material across the entire electrode thickness range, thereby achieving a low concentration gradient.

3). Critically, this mechanism is not a computational artifact. *Supplementary Fig. 1 experimentally validates our theoretical analysis,* demonstrating that low-concentration electrolytes enable more uniform prelithiation than high-concentration counterparts.

3. Furthermore, the prelithiation method used experimentally appears to involve fabricating a coin cell, performing prelithiation using a galvanostatic method, and then disassembling the cell to get a prelithiated electrode and to create a new one. (Even the detailed experimental procedure wasn't in the SI.) This method has been employed in numerous previous studies, making it neither novel nor practical.

Response: Thanks for your careful comments.

The work reports a prelithiation approach through direct contact between the pristine electrode and metallic Li, with carbonate solvents serving as the reaction mediums, which differs the electrochemical prelithiation as described by the reviewer. The operation does not need to involve fabricating a coin cell, performing prelithiation using a galvanostatic method, and then disassembling the cell to get a prelithiated electrode and to create a new one. We have shown 8 x 25 cm² prelithiated electrode in Fig. 5b, which is much larger than the size of coin cell. The detailed experimental procedures are provided in the supplementary materials (Line 36-40, Page 3, Supporting information).

Besides, this advanced prelithiation method with salt free medium avoids the drawbacks of traditional contact prelithiation methods that use high-concentration Li⁺ electrolyte medium, including material/electrode damage, significant Li salt residue, and complex post-treatment procedures.

4. If the prelithiation was performed through direct contact with lithium metal in the presence of an electrolyte, it remains unclear how a low-ionic-conductivity electrolyte could ensure uniform prelithiation on both the electrode surface and its inside. The rationale for this claim appears insufficient.

Response: Thanks for your careful comments. To address the reviewer's concerns, we conducted a thorough analysis both theoretically and experimentally.

a) We compute the steady-state Li⁺ concentration distribution within Si electrodes utilizing PyBaMM. Our results indicate that a reduction in the electrolyte's Li⁺ concentration diminishes the

concentration gradient across the electrode. We further introduce the prelithiation rate (R_p) as an index of the material's lithiation reaction rate, while defining the steady-state characteristic diffusion parameter (λ) of the porous electrode as a quantitative measure of the Li^+ supply from the electrolyte (Fig. 1d). Owing to the markedly reduced Li^+ concentration, there is a significant decrease in R_p . Moreover, the low Li^+ concentration medium, characterized by lower viscosity and an augmented Li^+ diffusion coefficient, exhibits an enhanced λ , thereby promoting a uniform distribution of Li^+ ions across the electrode depth.

b) We experimentally investigated the contact prelithiation of Si/C electrodes in electrolyte media with varying Li^+ concentrations (Supplementary Fig. 1). Our findings indicate that a lower Li^+ concentration invariably reduces the prelithiation reaction rate while enhancing its temporal uniformity. The electrode subjected to contact prelithiation in a salt-free medium (F-Si/C) was subsequently characterized and uniform electrode prelithiation was verified. As shown in Fig. 2a, the electrode maintained its original morphology and structure, and Fig. 2b confirms a uniform Li distribution. Furthermore, TEM measurements (Fig. 2e) revealed intact particles accompanied by a homogeneous SEI. Consequently, owing to the uniform prelithiation, full cells incorporating the F-Si/C electrode exhibit excellent electrochemical performance, as demonstrated in Figs. 2h and 3a. Collectively, these experimental outcomes substantiate the assertion that an electrolyte with low ionic conductivity can promote uniform prelithiation both at the electrode surface and within its interior.

In conclusion, both theoretically and experimentally, the results have demonstrated that low-ionic-conductivity electrolytes can ensure uniform prelithiation on both the electrode surface and its inside.

5. While Figure 5 presents a scale-up strategy, Figure 5a alone does not provide enough evidence to confirm whether a large-area electrode was indeed prelithiated using this approach.

Response: Thanks for your careful comments.

We fabricated an $8 \times 25 \text{ cm}^2$ electrode to demonstrate that a large-area electrode can be prelithiated using the as-explored prelithiation approach, as clearly depicted in Fig. 5b. To show the success of the prelithiation for large-area electrode, $8 \times 8 \text{ cm}^2$ Ah-level laminated pouch cell was constructed and the electrochemical performance was evaluated (Fig. 2g). The cell showed the high initial coulombic efficiency of 88.8%, which was much higher than 67.7% for that without prelithiation. Meanwhile, the cell delivered impressive cycling stability performance with a high capacity retention of 93% for 350 cycles at 1 C.

6. The use of FEC as a solvent is also not novel.

Response: Thanks for your careful comments.

Although FEC has been widely used as a solvent in LIB electrolytes, a comprehensive review of the literature has led us to conclude that FEC is employed as a prelithiation medium in contact prelithiation for the first time. Importantly, we reveal the underlying mechanism, that FEC can react with metallic Li to generate ultralow-concentration Li^+ ions (at the millimolar L^{-1} level) while enabling capability of ionic conductivity.

7. Above all, the methods proposed and used in this study are not novel at all, so I don't believe this paper is suitable for publication in the highly prestigious Nature Communications.

Response: Thanks for your careful comments.

Conceptually, this work reveals fundamental understanding of regulation of prelithiation reaction to improve the electrochemical performance of high-energy-density LIBs. Technologically, this work presents a salt-free medium (cyclic carbonate solvent) that can homogenize electrode contact prelithiation, synchronously achieving efficient Li compensation and fast electrochemical reaction kinetics of the anode. As a result, it enables the development of LIBs with both fast-charging capability and high energy density. Moreover, this specific advanced prelithiation method circumvents the drawbacks of traditional contact prelithiation method using high Li⁺ concentration electrolyte mediums, including material/electrode damage, abundant Li salt residue and the complex post-processing procedures. Based on the above consideration, we are confident that this work is an important breakthrough in the existing LIBs and will greatly interest readers, and reaches the very high standard of *Nature Communications*.

Reviewer 4:

Comments:

This paper reported the uniform distribution of Li⁺ ion across the depth of electrode using various cyclic carbonate mediums during prelithiation, to boost the energy density, cycling performance improvements as well as fast charging capabilities. The author applied this method with various types of anode materials and paired with different cathodes. Moreover, comprehensive and thorough characterization techniques were implemented to elucidate the working mechanism. However, I have a few suggestions for the improvement and hope the author can find it helpful.

Response: Thank you very much for your careful and positive comments, which help to improve the quality of our work.

1. Definition of prelithiation degree and how to control? What is the prelithiation setup looks like? Normally, during prelithiation, the electron will roll up due to the volume change of Si, is there any pressure applied to the electrode during prelithiation in this study? If using roll-to-roll method as illustrated in Fig 5a, during long prelithiation process (~ 300 mins), how to prevent solvent evaporation?

Response: Thanks for your careful comments.

1). The degree of prelithiation is defined as the ratio of the difference in discharge capacity between fresh Si/C and prelithiated Si/C to the discharge capacity of fresh Si/C. This calculation method was consistent with that described in reference [Advanced Energy Materials **13**, 2203256 (2023)]. The prelithiation degree is controlled by adjusting the prelithiation time for a given cyclic carbonate such as FEC.

2). Prelithiation setup: for lab-scale fabrication, two glass plates are employed to secure the electrode and Li metal in intimate contact. This process is conducted inside an argon-filled glove box, where the oxygen and water concentrations are maintained at ≤ 0.5 ppm and ≤ 0.1 ppm, respectively.

3). During prelithiation, the two glass plates were subjected to a pressure of approximately 10 kPa, which effectively suppressed electrode rolling. Moreover, the overall degree of prelithiation in the

electrodes was kept minimal (25% for the F-Si/C), thereby preventing significant strain that could lead to rolling.

4). As illustrated in the schematic diagram of Fig. 5a, for roll-to-roll applications, the holder (e.g., two glass plates) secures the designated reaction area, thereby establishing a relatively sealed environment that substantially reduces solvent evaporation.

2. Line 162 Augur electron spectroscopy (AES), what is the approx. depth of electrode in nanometer? There is bigger fluctuation of O species, what these contents could be?

Response: Thanks for your careful comments.

1). The detection depth of AES is approximately 2 nm. In this study, measurements were performed by selecting specific points along the cross-section of the electrode in Fig. 2a.

2). For E-Si/C, both oxygen and lithium exhibit pronounced fluctuations, indicative of uneven prelithiation. The oxygen originates from multiple sources, including both the active material and the post-reaction interface. Given the considerable and non-uniform thickness of the E-Si/C interface combined with the presence of organic components, oxygen fluctuations are markedly elevated.

3. Line 185-188, how did you normalize the wt% number for LiF and total inorganic species, and which signature peaks are considered as inorganic species?

Response: Thanks for your careful comments.

The calculation method for determining the mass fraction is outlined as follows:

(1) Initially, the atomic content of a compound is determined by multiplying the elemental atomic ratio by the area ratio of the corresponding characteristic peak. For example, in F-Si/C, the atomic ratio of fluorine is 27.3%, and the area ratio of the LiF peak in the high-resolution F 1s spectrum is 92.5%, yielding an atomic content for LiF of 25.2%.

(2) Next, the molecular mass of the compound is computed by multiplying its atomic content by its relative molecular mass. Specifically, with a relative molecular mass of 25.9 for LiF, the molecular mass is calculated to be 6.5.

(3) Subsequently, the total mass of all elements is calculated as the sum of the products of the atomic content and the relative atomic mass for each constituent. In F-Si/C, the combined mass of all elements (Li, C, F, O, Si, P) is 13.0.

(4) Finally, the mass fraction of the compound is obtained by dividing the molecular mass of the compound by the total mass of all elements. For LiF, this calculation yields a mass fraction of $6.5 / 13.0$, or 50.5%.

Based on this methodology, the mass fractions of Li_2O and Li_2CO_3 can similarly be determined, with their combined mass fractions, together with that of LiF, constituting the total inorganic mass fraction. The characteristic peaks used for this analysis are as follows: LiF at 685 eV in the high-resolution F 1s spectrum, Li_2CO_3 at 290 eV in the high-resolution C 1s spectrum, and Li_2O at 54 eV in the high-resolution Li 1s spectrum (see Supplementary Fig. 11).

The following discussions were added to the Supporting information:

" LiF, Li_2O and Li_2CO_3 species constitute the total inorganic components, and their corresponding characteristic peaks are as follows: LiF at 685 eV in the high-resolution F 1s

spectrum, Li_2CO_3 at 290 eV in the high-resolution C 1s spectrum, and Li_2O at 54 eV in the high-resolution Li 1s spectrum. " (Page 18, Supporting information)

4. What is the structure of pristine Si/C without any treatment? TEM data should be provided for comparison. Any lithiated Si/C compounds were identified in the bulk region after prelithiation?

Response: Thanks for your careful comments. To address the reviewer's concerns, we conducted a comprehensive TEM analysis of the Si/C and F-Si/C samples.

1). Figure R11 presented the TEM data of pristine Si/C without any additional treatment. The TEM micrograph revealed that the Si/C particles were uniformly encompassed by an ultrathin carbon layer, approximately 5 nm in thickness. The cores of the particles displayed well-defined lattice fringes corresponding to the (111) crystal plane of Si, with an interplanar spacing of 0.31 nm. Thus, the pristine Si/C structure consisted of Si particles encapsulated by a thin carbon layer.

2). TEM analysis was employed to examine the structure of F-Si/C (Figure R12). The TEM micrograph revealed a coherent surface layer, approximately 11 nm thick, on the F-Si/C particles. Within this layer, nanodomains of LiF, Li_2CO_3 , and Li_2O —approximately 3 nm in size—were embedded. This observation indicated the formation of an inorganic-rich interfacial layer during the FEC-mediated contact prelithiation process. It was noteworthy, however, that the overall degree of prelithiation is low (25%), as no lithiated Si/C compounds were identified in the bulk region. XRD was conducted to assess the crystallinity of both F-Si/C and pristine Si/C (Figure R13). The F-Si/C electrode exhibited two prominent peaks at 28.4° and 26.3° , corresponding to the (111) plane of silicon and the (002) plane of graphite, respectively. The similarity in characteristic peaks for both the F-Si/C and pristine Si/C electrodes suggested that the FEC-mediated prelithiation, given its low prelithiation degree, predominantly occurred on the surface of the active material rather than penetrating its internal structure.

Figure R11. High-resolution TEM image for the Si/C particle.

Figure R12. High-resolution TEM image for the F-Si/C.

Figure R13. XRD patterns of the Si/C electrode and F-Si/C electrodes.

The following discussions were added to the Supporting information:

" The prelithiation degree of prelithiated Si/C electrodes was ~25%. " (Page 17, Supporting information)

Response to reviewers' comments:

(Blue italic: Reviewers' remarks; Black type: Our response)

Reviewer 3:

Comments:

While I acknowledge that the results presented in this manuscript are technically impressive, I initially believed that the work lacked sufficient novelty to merit publication in a high-impact journal such as Nature Communications. However, in light of the editor's suggestion and the authors' responses, I have re-evaluated the manuscript and offer the following comments and questions.

Response: Thank you very much for your careful and helpful evaluation of our work. Please see the detailed responses to the following comments.

1. As the authors are aware, electrodes, such as those shown in Figures 1a and 1b, are not flat but inherently porous. In such porous electrode architectures, when the proposed direct contact prelithiation method is applied, lithium metal comes into immediate contact with the topmost silicon particles. This results in extremely rapid, uncontrolled prelithiation at the surface, while the interior silicon (near the current collector) remains unreacted. Based on the manuscript, it appears that the simulation model does not account for this direct Li-Si contact at the top surface. Even if the authors mitigate concentration gradients by using a low-concentration electrolyte, they cannot prevent over-lithiation of silicon particles that are directly touching lithium metal. Therefore, I argue that depth-wise nonuniformity in prelithiation is inevitable with this direct contact method, regardless of electrolyte engineering. From the standpoint of achieving uniform prelithiation across all particles in the electrode, I believe that this strategy is not the best method. If the authors are proposing a resting step (or self-discharge between the Si particles having different SOC) as a solution to this nonuniformity, I would like to ask how this is fundamentally different from what occurs under lithium metal contact with conventional electrolytes.

Response: Thanks for your careful comments. We have thoroughly considered these comments and concluded that the primary concern pertains to the uniformity of the cyclic carbonate-mediated prelithiation. One of the principal contributions of this work is the elucidation of the relationship between the Li^+ concentration in the medium and the uniformity of prelithiation degree across the electrode thickness of contact prelithiation. Theoretical simulation and experimental investigation have been conducted to reveal the depth-dependent uniformity in prelithiation with mediums with different Li^+ concentrations (see details at Paragraph2 Page 5, Paragraph1, Page 6 in the main text).

For your convenience, these contents have also been summarized as follows:

- 1) The simulation results employing PyBaMM (Fig. 1c and 1e) demonstrates that lowering Li^+ concentration of the medium can reduce the prelithiation degree gradient across the depth of the electrode, leading to more homogeneous prelithiation at electrode level. While Fig. 1 presents a schematic representation of a dense electrode configuration for easy understanding, the simulation results are applicable to both porous electrodes and dense film electrodes.
- 2) The Li contents of the prelithiated Si/C electrodes at different electrode depths were examined using Auger electron spectroscopy (E-Si/C with 1M LiPF_6 medium and F-Si/C with FEC medium, both with 25% prelithiation degree, Fig. 2b and Supplementary Fig. 5). The Li contents for the F-Si/C electrode were 27% at 10% depth of electrode and 23% at 70% depth of electrode, respectively. In contrast, the corresponding Li contents on the E-Si/C electrode were 64% and 16% at the two

equal depths of the electrode, respectively.

Thus, the employment of ultralow Li^+ ion concentration medium (10 mM level Li^+ in cyclic carbonate medium, FEC) significantly improve the homogeneity of prelithiated electrode across the electrode thickness in comparison to that using regular electrolyte medium (1 M level Li^+ in regular electrolyte).

The reason for excluding the direct Li-Si contact in the simulation model lies in that this work mainly focuses on the contact prelithiation reaction with liquid mediums with different Li^+ concentrations. The solid-state reaction between metallic Li and Si at ambient condition is inherently slow [Advanced Functional Materials **32**, 2201455 (2022)].

2. The authors replied that using FEC in the method of direct contact prelithiation is novel. Even if that is technically true (which I am not convinced of), FEC has long been known as a No.1. critical additive for forming stable SEI layers on silicon or silicon-carbon composite anodes. Thus, it is difficult to regard the use of FEC itself as a meaningful novelty.

Response: Thanks for your careful comments.

The key finding of this study is that cyclic carbonate solvents can effectively serve as mediums to promote a uniform prelithiation for electrodes. In this study, the main function of the cyclic carbonate solvents lies in regulating the uniformity of prelithiation reaction. While FEC is well-known as a critical additive for forming a stable SEI layer on Si or Si/C composite anodes, this study presents the first reported investigation into its role—along with other cyclic carbonate solvents—in promoting uniform electrode prelithiation. This finding constitutes a major innovation of this work.

3. The authors demonstrate excellent performance using various pouch cells, including a 1 Ah-level cell. While fabricating such a large cell at the university level is certainly not trivial, what ultimately matters is the electrodes' areal loading level. If the reported high capacity is primarily achieved by stacking many low-loading electrodes, it is not a practical approach and does not represent a high energy-density design. According to the table, the cathode loading appears to be $\sim 2 \text{ mAh/cm}^2$, which is not considered high. In this case, the anode thickness must be extremely thin due to the high specific capacity of silicon. Could this be why the uniformity issue I raised in Comment 1 was not significant in your experiments? In other words, did the thinness of the electrode simply mask the depth-wise nonuniformity in prelithiation? If the proposed prelithiation method is truly practical and scalable, its effectiveness should be demonstrated in electrodes with at least 4 mAh/cm^2 areal capacity, and the full-cell tests should be conducted accordingly. The important point is not just achieving high Ah capacity, but doing so with industrially relevant electrode specifications.

Response: Thanks for your careful comments.

As discussed in the response to Question 1, the Li^+ concentration in the medium significantly impacts the uniformity of electrode prelithiation. Mediums with high Li^+ concentrations tend to result in a large prelithiation gradient along the electrode thickness, whereas low Li^+ -concentration mediums can significantly reduce this gradient, thereby improving the uniformity of prelithiation across the electrode thickness. In the experiments, $\sim 30 \mu\text{m}$ thick-electrodes with an areal capacity of $\sim 2 \text{ mAh cm}^{-2}$ were used. The Li contents of the prelithiated Si/C electrodes at different electrode depths were examined using Auger electron spectroscopy (E-Si/C with 1M LiPF_6 medium and F-Si/C with FEC medium, both with 25% prelithiation degree, Fig. 2b and Supplementary Fig. 5). The Li contents for the F-Si/C electrode were 27% at 10% depth of electrode and 23% at 70% depth of

electrode, respectively. In contrast, the corresponding Li contents on the E-Si/C electrode were 64% and 16% at the two equal depths of the electrode, respectively. Thus, the FEC medium enables much improved uniformity of prelithiation across the electrode thickness in comparison to the electrolyte.

According to calculation of the relative Li^+ concentration (Figure R1), it is revealed that for thicker electrodes (e.g., 30-100 μm), the increased ion diffusion distance further amplifies the improvement in prelithiation uniformity observed with low Li^+ concentration medium. It is important to emphasize that while low Li^+ concentration medium help improve prelithiation uniformity along the electrode thickness, this does not imply they achieve absolute prelithiation uniformity.

Figure R1. The relationship between the relative concentration at the bottom of the electrode and the thickness of the electrode after prelithiation employing mediums with different c_{Li^+} using PyBaMM package. The electrode thickness is extended to 100 μm based on the result of Fig. 1c.